# Coupling Category Alignment for Graph Domain Adaptation

## Abstract

Graph domain adaptation (GDA), which transfers knowledge from a labeled source domain to an unlabeled target graph domain, attracts considerable attention in numerous fields. However, existing methods commonly employ message-passing neural networks (MPNNs) to learn domain-invariant representations by aligning the entire domain distribution, inadvertently neglecting category-level distribution alignment and potentially causing category confusion. To address the problem, we propose an effective framework named **Co**upling **C**ategory **A**lignment (CoCA) for GDA, which effectively addresses the category alignment issue with theoretical guarantees. CoCA incorporates a graph convolutional network branch and a graph kernel network branch, which explore graph topology in implicit and explicit manners. To mitigate category-level domain shifts, we leverage knowledge from both branches, iteratively filtering highly reliable samples from the target domain using one branch and fine-tuning the other accordingly. Furthermore, with these reliable target domain samples, we incorporate the coupled branches into a holistic contrastive learning framework. This framework includes multi-view contrastive learning to ensure consistent representations across the dual branches, as well as cross-domain contrastive learning to achieve category-level domain consistency. Theoretically, we establish a sharper generalization bound, which ensures the effectiveness of category alignment. Extensive experiments on benchmark datasets validate the superiority of the proposed CoCA compared with baselines.

## 1 Introduction

As a crucial problem in graph classification (Lin et al., 2023; Luo et al., 2023), Graph Domain Adaptation (GDA) has received substantial interest, particularly in the fields of temporally-evolved social analysis (Wang et al., 2021), molecular biology (You et al., 2022b; Zhu et al., 2023; Yin et al., 2023), and protein-protein interaction networks (Cho et al., 2016). GDA transfers graph representations learned from the source domain to the target domain, which is necessary in many applications. Domain adaptive learning is inherently challenging due to the distribution shift between source and target domains (i.e., $\mathbb{P}_S(G, Y) \neq \mathbb{P}_T(G, Y)$). This challenge is further amplified when handling graph-structured data, which often represent abstractions of varying natures (You et al., 2022a).

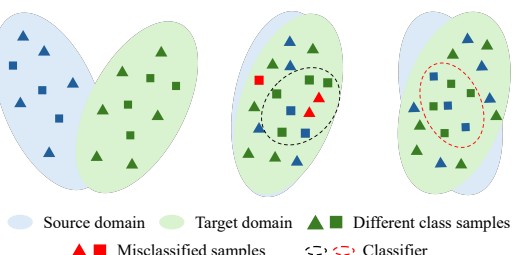

Source domain    Target domain    ▲ ■ Different class samples
▲ ■ Misclassified samples    Classifier

Figure 1: Left: The source and target graph domains. Middle: GDA methods that align the entire source and target domains, potentially confuse category distribution (see the red triangles and squares). Right: The proposed method, which aligns category-level distributions, alleviates the category-agnostic issue.

Currently, various GDA methods have been proposed (Yin et al., 2022; 2023; You et al., 2022b; Zhu et al., 2023) by combining domain adaptation techniques with graphs. They usually assume the distribution invariance is limited (Garg et al., 2020; Verma & Zhang, 2019) and directly employ adversarial training to align source and target distribution (Zhang et al., 2019b; Wu et al., 2020a). However, the classifier still tends to favor source domain features and makes incorrect predictions

on the target domain due to category-agnostic feature alignment (Zhang et al., 2019a). To solve the issue and efficiently design the GDA framework, we still need to address the following challenges: (i) *How to fully exploit the features of the source and target domain for representation learning.* Previous approaches typically employ the MPNNs to capture implicit topological semantics. However, the absence of labels for the target domain poses challenges in obtaining sufficient topological semantics. (ii) *How to effectively align category-level distribution.* While there has been progress in matching the marginal distributions between two domains, they may not efficiently align the category distribution, leading to a degradation in classification performance. Certain methods have attempted to acquire pseudo-labels for the target graphs in target domain training (Yin et al., 2023; Yehudai et al., 2021; Ding et al., 2021; Zhu et al., 2023), they are vulnerable to bias in cases of significant domain shift, leading to error accumulation in subsequent optimization. (iii) *How to design the GDA framework with the grounded theoretical foundation.* Theoretically, the generic domain adaptation (DA) bound is not specific to graph data and models (You et al., 2023). However, we can still design a more precise model tailored for graphs with the theoretical guarantee.

To tackle these challenges, we propose a framework named **Co**pling **C**ateg**o**ry **A**lignment (CoCA) for unsupervised domain adaptive graph classification. Specifically, to fully exploit the features of both source and target graphs, CoCA incorporates an MPNN branch and a shortest path aggregation branch. The MPNN branch leverages neighborhood aggregation to implicitly learn topological semantics, while the shortest path aggregation branch generates paths for each node and utilizes position encoding to extract informative graph-level semantics. This shortest path aggregation branch provides explicit high-order structural semantics, serving as a complementary enhancement. To collaborate knowledge from two branches, we jointly train the branches by iteratively filtering highly reliable samples from the target domain using one branch and fine-tuning the other branch accordingly. Specifically, with the dual pre-trained branches, CoCA first fine-tunes the shortest path branch with the highly reliable samples filtered from the target domain with the MPNN branch and then optimizes the MPNN branch with the filtered target domain samples labeled by the shortest path branch. Theoretically, the interactive optimization of one branch with the support of the other one would gradually mitigate the category-level distribution shift. Furthermore, we embed the iterative learning process into a holistic contrastive learning framework, incorporating cross-domain contrastive learning to achieve category-level domain consistency, alongside multi-views contrastive learning to ensure consistent representations between branches. Overall, our approach emphasizes a unique focus on achieving category-level domain alignment. Specifically, our methodology is centered on Coupling Category Alignment (CoCA), which systematically iterates between branches to identify and select reliable samples. This process facilitates cross-branch adjustments, effectively mitigating potential domain shifts in an unsupervised manner. By iteratively refining the alignment process, our approach enhances the model's ability to achieve category-level alignment, supported by solid theoretical foundations, distinguishing it from prior methods. Extensive experiments conducted on various datasets with domain shifts for graph classification demonstrate that the proposed CoCA outperforms state-of-the-art baselines.

In summary, the main contributions can be summarized as three-fold: (1) *Problem Formulation*: We present a novel problem in graph domain adaptation, which highlights the discrepancy in the distribution of graph categories between the source and target domains, posing significant challenges for accurate graph classification across domains. (2) *Methodology*: We propose a framework named CoCA, which utilizes two branches to explore structural semantics and integrates them into a category-level domain-invariant model. We provide theoretical proof demonstrating that CoCA is specifically designed to more accurately address the challenges of the graph domain. (3) *Experiments*: Extensive experiments conducted on various domain shift datasets for graph classification demonstrate the effectiveness of the proposed CoCA.

## 2 RELATED WORK

**Graph Classification.** GNNs (Kipf & Welling, 2017b) have shown exceptional performance across a wide range of graph-based machine learning tasks, such as node classification (Rong et al., 2020; Hu et al., 2019; Kipf & Welling, 2017b), graph classification (Fan et al., 2019; Song et al., 2016; Liao et al., 2021; Wu et al., 2020b) and link prediction (Zhang & Chen, 2018; Cai et al., 2021). The most prevalent GNNs follow the message-passing paradigm, which aggregates the neighbors for node update and applies graph pooling for graph representation. Nevertheless, MPNNs have

limited capacity to capture high-order topological structures, such as paths and motifs (Du et al., 2019; Michel et al., 2023; Ju et al., 2022; Qiu et al., 2021). Therefore, numerous graph kernel methods have emerged to overcome this flaw (Long et al., 2021; Cosmo et al., 2021; Wang et al., 2022). However, these approaches typically require an ample supply of labeled annotations (Yin et al., 2022; 2023) while this work delves into the realm of unsupervised graph domain adaptation and introduces a novel approach CoCA to tackle this challenge.

**Unsupervised Domain Adaptation.** Unsupervised domain adaptation is to learn domain-invariant representations that enable the transfer of a model from a source domain with abundant labels to a target domain with a scarcity of labels (Ma et al., 2021; Singh, 2021; Feng et al., 2023; Kouw & Loog, 2019). The majority of technical routes can be broadly categorized into domain discrepancy-based methods and adversarial approaches. The former methods typically incorporate different distribution metrics like maximal mean discrepancy (Saito et al., 2018) and Wasserstein distance (Shen et al., 2018) to measure the discrepancy between different domains. Conversely, adversarial approaches involve a domain discriminator that is fused to implicitly reduce the domain discrepancy. However, these methods typically concentrate on Euclidean data such as images and texts, while graph domain adaptation has not been extensively explored. In this work, we explores graph semantics by utilizing dual perturbation branches for effective graph domain adaptation.

**Graph Domain Adaption.** Due to the potential economic value, graph domain adaptation (Lin et al., 2023; Wu et al., 2022b; Luo et al., 2023) is a crucial problem in the fields of social analysis and molecular biology (You et al., 2022b; Zhu et al., 2023). Existing methods mainly focus on how to transfer information from source graphs to unlabeled target graphs to learn effective node-level (Wu et al., 2022a; Zhu et al., 2021; Dai et al., 2022; Guo et al., 2022) and graph-level (Yin et al., 2023; Yehudai et al., 2021; Ding et al., 2021; Yang et al., 2020) representation. However, these approaches commonly merge GNNs with domain alignment (Luo et al., 2023) techniques, which overlook the alignment of category distributions in the presence of label scarcity and domain shift, consequently leading to a deterioration in classification performance. Towards this end, CoCA couples the dual branch in a variational optimization framework to address the issue.

## 3 PRELIMINARY

**Problem Setup.** Denote a graph as $G = (V, E, \mathbf{X})$ with the node set $V$, the edge set $E$, and the node attribute matrix $\mathbf{X} \in \mathbb{R}^{|V| \times F}$ with $F$ denotes the attribute dimension and $|V|$ denotes the number of nodes. The labeled source domain is denoted as $\mathcal{D}^s = \{(G_i^s, y_i^s)\}_{i=1}^{N_s}$, where $y_i^s$ denotes the labels of $G_i^s$. The unlabeled target domain is $\mathcal{D}^t = \{G_j^t\}_{j=1}^{N_t}$, where $N^s$ and $N^t$ denote the number of source graphs and target graphs. Both domains share the same label space $\mathcal{Y}$, but have different distributions in the graph space. Our objective is to train a model using both labeled source graphs and unlabeled target graphs to achieve superior performance in the target domain.

**DA Bound for Graph.** Applying GDA with optimal transport (OT), if the covariate shift holds on representations that $\mathbb{P}_S(Y|Z) = \mathbb{P}_T(Y|Z)$, the target risk $\epsilon_T(h, \hat{h})$ is bounded with the theorem:

**Theorem 1** *(Redko et al., 2017; Shen et al., 2018) Suppose the learned discriminator $g$ is $C_g$-Lipschitz where the Lipschitz norm $||g||_{Lip} = \max_{Z_1, Z_2} \frac{|g(Z_1) - g(Z_2)|}{\rho(Z_1, Z_2)} = C_g$ holds for some distance function $\rho$ (Euclidean distance here). Let $\mathcal{H} := \{g : \mathcal{Z} \to \mathcal{Y}\}$ be the set of bounded real-valued functions with the pseudo-dimension $Pdim(\mathcal{H}) = d$ that $g \in \mathcal{H}$, with probability at least $1 - \delta$ the following inequality holds:*

$$\epsilon_T(g, \hat{g}) \leq \hat{\epsilon}_S(g, \hat{g}) + \sqrt{\frac{4d}{N_S} \log(\frac{eN_S}{d}) + \frac{1}{N_S} \log(\frac{1}{\delta})} + 2C_g W_1(\mathbb{P}_S(Z), \mathbb{P}_T(Z)) + \omega,$$

*where $\omega = \min_{||g||_{Lip} \leq C_g} \{\epsilon_S(g, \hat{g}) + \epsilon_T(g, \hat{g})\}$ denotes the model discriminative ability, and the first Wasserstein distance is defined as (Villani et al., 2009): $W_1(\mathbb{P}, \mathbb{Q}) = \sup_{||g||_{Lip} \leq 1} \{\mathbb{E}_{\mathbb{P}_S(Z)} g(Z) - \mathbb{E}_{\mathbb{P}_T(Z)} g(Z)\}$.*

**Theorem 2** *(You et al., 2023) Assuming that the learned discriminator is $C_g$-Lipschitz continuous as described in Theorem 1, and the graph feature extractor $f$ (also referred to as GNN) is $C_f$-Lipschitz that $||f||_{Lip} = \max_{G_1, G_2} \frac{||f(G_1) - f(G_2)||_2}{\eta(G_1, G_2)} = C_f$ for some graph distance measure $\eta$.*

Figure 2: An overview of the proposed CoCA. CoCA contains a message passing branch and a shortest path aggregation branch. To align category-level distribution, we alternatively optimize each branch with highly dependable pseudo-labels learned from the other branch. CoCA incorporates the learning process in a multi-view and cross-domain contrastive learning framework.

*Let $\mathcal{H} := \{h : \mathcal{G} \rightarrow \mathcal{Y}\}$ be the set of bounded real-valued functions with the pseudo-dimension $Pdim(\mathcal{H}) = d$ that $h = g \circ f \in \mathcal{H}$, with probability at least $1 - \delta$ the following inequality holds:*

$$\epsilon_T(h, \hat{h}) \leq \hat{\epsilon}_S(h, \hat{h}) + \sqrt{\frac{4d}{N_S} \log(\frac{eN_S}{d}) + \frac{1}{N_S} \log(\frac{1}{\delta})} + 2C_f C_g W_1(\mathbb{P}_S(G), \mathbb{P}_T(G)) + \omega,$$

*where the (empirical) source and target risks are $\hat{\epsilon}_S(h, \hat{h}) = \frac{1}{N_S} \sum_{n=1}^{N_S} |h(G_n) - \hat{h}(G_n)|$ and $\epsilon_T(h, \hat{h}) = \mathbb{E}_{\mathbb{P}_T(G)}\{|h(G) - \hat{h}(G)|\}$, respectively, where $\hat{h} : \mathcal{G} \rightarrow \mathcal{Y}$ is the labeling function for graphs and $\omega = \min_{||g||_{Lip} \leq C_g, ||f||_{Lip} \leq C_f} \{\epsilon_S(h, \hat{h}) + \epsilon_T(h, \hat{h})\}$.*

The comprehensive justification of the OT-based graph domain adaptation bound demonstrates that the generalization gap relies on both the domain divergence $2C_f C_g W_1(\mathbb{P}_S(G), \mathbb{P}_T(G))$ and model discriminability $\omega$.

## 4 METHODOLOGY

This work studies the unsupervised GDA problem (Yin et al., 2023; Yehudai et al., 2021) and proposes a new approach CoCA (see Figure 1). CoCA consists of two parts, the **dual graph branch** explores semantics from implicit and explicit perspectives; the **branch coupling module** interactively optimizes one branch with highly reliable sample filtered from the other branch to minimize the category distribution discrepancy. CoCA incorporates the iterative process into a learning framework and theoretical proof the designed method is more precisely tailored for the graph domain.

### 4.1 DUAL BRANCH FOR SEMANTICS MINING

Current graph transfer learning methods (Sun et al., 2022; Lin et al., 2023; Wu et al., 2022b; Lee et al., 2017) typically rely on MPNNs to implicitly capture topological semantics through neighborhood aggregation for transfer learning. However, these approaches may be suffered under domain shift. To address this issue, we introduce a dual-branch architecture for graph representation learning, comprising a MPNNs branch for implicit topological semantics and a shortest path aggregation branch for explicit topological semantics derived from high-order structures.

**Message Passing Branch.** MPNNs extract graph semantics by aggregating neighborhood nodes to update each central node representation. We update the representation of node $u$ at layer $l$ in the massage passing branch $f_\theta^{MP}(\cdot)$ and summarize the node representations into graph-level as:

$$\mathbf{h}_u^l = \mathcal{C}_{MP}^l\left(\mathbf{h}_u^{l-1}, \mathcal{A}_{MP}^l\left(\{\mathbf{h}_v^{l-1}\}_{v \in \mathcal{N}(u)}\right)\right), \quad \mathbf{z}^{MP} = f_\theta^{MP}(G) = \text{READOUT}\left(\{\mathbf{h}_u^L\}_{u \in V}\right),$$

where $\mathcal{N}(u)$ is the neighbours of node $u$. $\mathcal{C}_{MP}^l$ and $\mathcal{A}_{MP}^l$ are combination and aggregation functions at layer $l$, and READOUT is the pooling function. In this way, the message passing branch learns the topological structure in an implicit manner under label supervision.

**Shortest Path Aggregation Branch.** However, the message passing branch merely extracts topological structural semantics in an implicit manner, which would be challenged under the circumstance of domain shift. Considering an alternative technical route, graph kernels (Borgwardt & Kriegel, 2005; Shervashidze et al., 2011; Gao et al., 2021) are capable of explicitly extracting high-order semantics. We introduce a shortest path aggregation branch that generates various shortest paths from local substructures to extract high-order semantics into graph-level representations. Thus, the representations alleviate the impact of structural shift across domains.

In particular, denote $\mathcal{N}_k(u)$ as the set of nodes reachable from $u$ through a shortest path of length $k$, and the representation of node $u$ can be updated with $\mathcal{N}_1(u) \cup \cdots \cup \mathcal{N}_k(u) \cup \cdots \cup \mathcal{N}_K(u)$, $K$ is the hyperparameter of largest length. We update the nodes on different path length $k$:

$$\mathbf{m}_{u,\mathcal{N}_k(u)}^l = \mathcal{C}_{SP}^l \left( \hat{\mathbf{m}}_u^{l-1}, \mathcal{A}_{SP}^l \left( \{ \hat{\mathbf{m}}_v^{l-1} \}_{v \in \mathcal{N}_k(u)} \right) \right),$$

where $\mathcal{C}_{SP}^l$ and $\mathcal{A}_{SP}^l$ denotes combination and aggregation operators at layer $l$ on shortest path branch. Thus, we obtain the embeddings from different path length, i.e., $\left\{ \mathbf{m}_{u,\mathcal{N}_1(u)}^l, \cdots, \mathbf{m}_{u,\mathcal{N}_K(u)}^l \right\}$, and update the representation of $u$ as follows:

$$\boldsymbol{\alpha}^l = Atten \left( ||_{k=1}^K \mathbf{m}_{u,\mathcal{N}_k(u)}^l \right), \quad \mathbf{m}_u^l = MLP \left( (1+\epsilon)\mathbf{m}_u^{l-1} + \sum_{k=1}^K \alpha_k^l \mathbf{m}_{u,\mathcal{N}_k(u)}^l \right),$$

where $\epsilon \in \mathbb{R}$, $Atten$ is the self-attention mechanism, $||$ denotes the concatenation operation, and $||_{k=1}^K \mathbf{m}_{u,\mathcal{N}_k(u)}^l \in \mathbb{R}^{K \times d'}$, $d'$ is the feature dimension of $\mathbf{m}_{u,\mathcal{N}_k(u)}^l$. $MLP$ is the fully connected layer. After stacking $L$ layers, we take the average of all nodes into a graph-level representation:

$$\mathbf{z}^{SP} = f_\phi^{SP}(G) = \text{READOUT} \left( \{ \mathbf{m}_u^L \}_{u \in V} \right). \tag{1}$$

The READOUT function is similar to Eq. 1. The shortest path aggregation branch acquires topological semantics by focusing on paths of varying lengths, which help mitigate the impact of structural domain shifts, such as differences in density and graph size.

## 4.2 Branch Coupling for Category Alignment

Recent work attempted to obtain target graph pseudo-labels for training (Yehudai et al., 2021; Ding et al., 2021; Zhu et al., 2023). However, due to discrepancies in the category distribution, they may suffer from error accumulation during subsequent optimization. To address this issue, we cleverly utilize the characteristics of the dual branch to obtain pseudo-labels and mitigate error accumulation.

Considering the message passing branch (MP branch) and the shortest path branch (SP branch), our objective is to identify highly dependable pseudo-labels in the target domain and integrate them with the source domain to fine-tune the model. In this way, we can efficiently align the category distribution. Nevertheless, with the challenges of category discrepancy and error accumulation, we cannot achieve satisfactory pseudo-labels in a signal branch. To address the issue, we introduce the distribution $p_\theta(\cdot)$ and $q_\phi(\cdot)$ for the MP and SP branches, and aim to align category distribution with the source and target labels $Y^s$ and $Y^t$. Specifically, in the MP branch, we filter the highly dependable pseudo-labels with the threshold $\zeta$, and use those samples to help fine-tune the SP branch:

$$\mathcal{L}_1 = \mathbb{E}_{p_\theta(\hat{y}_i^t | G^s, G^t, Y^s) > \zeta} \left[ \log q_\phi \left( \hat{y}_i^t \mid G_i^t \right) \right] + \sum_{i=1}^{N_s} \log q_\phi \left( Y_i^s \mid G_i^s \right), \tag{2}$$

where $\hat{y}^t$ is the target graph pseudo-labels filtered from the MP branch. Similarly, we utilize the target samples filtered from the SP branch to support the fine-tune of the MP branch:

$$\mathcal{L}_2 = \mathbb{E}_{q_\phi(\hat{y}_i^t | G^s, G^t, Y^s) > \zeta} \left[ \log p_\theta \left( \hat{y}_i^t \mid G_i^t \right) \right] + \sum_{i=1}^{N_s} \log p_\theta(Y_i^s \mid G_i^s). \tag{3}$$

The interactive optimization of the MP and SP branches offers two advantages. First, by incorporating highly confident target pseudo-labels into source domain training, we can effectively align the category distribution. Second, the pseudo-labels filtered from the other branch help mitigate the error accumulation issue caused by the single model.

**Theoretical Analysis.** Intuitively, incorporating training samples from target and source domains would effectively align the category distribution between domains. However, the theoretical basis for why iterative fine-tuning achieves category alignment still requires further investigation. Additionally, the graph category alignment bound remains agnostic. To address this, we present Theorem 3, which proves that the iterative learning process maximizes the Evidence Lower Bound (ELBO). Furthermore, Theorem 4 demonstrates that employing a category distribution alignment module results in a lower bound on the empirical risk in the target domain compared to without this module.

**Theorem 3** *The iterative learning process (i.e., Eq. 2 and 3) follows the optimization of maximizing the Evidence Lower Bound (ELBO):*

$$\log p_\theta \left(Y^s \mid G^s, G^t\right) \geq \mathbb{E}_{q_\phi(Y^t|G^t)} \left[\log p_\theta \left(Y^s, Y^t \mid G^s, G^t\right) - \log q_\phi \left(Y^t \mid G^t\right)\right]. \tag{4}$$

**Theorem 4** *Under the assumption of Theorem 2, we further assume that there exists a small amount of high dependable i.i.d. samples with pseudo labels $\{(G_n, Y_n)\}_{n=1}^{N_T'}$ filter from the target distribution $\mathbb{P}_T(G, Y)$ ($N_T' \ll N_S$) and bring in the conditional shift assumption that domains have different labeling function $\hat{h}_S \neq \hat{h}_T$ and $\max_{G_1,G_2} \frac{|\hat{h}_D(G_1) - \hat{h}_D(G_2)|}{\eta(G_1,G_2)} = C_h \leq C_f C_g (D \in \{S,T\})$ for some constant $C_h$ and distance measure $\eta$. Let $\mathcal{H} := \{h : \mathcal{G} \to \mathcal{Y}\}$ be the set of bounded real-valued functions with the pseudo-dimension $Pdim(\mathcal{H}) = d$, with probability at least $1 - \delta$ the following inequality holds:*

$$\epsilon_T(h, \hat{h}_T) \leq \frac{N_T'}{N_S + N_T'} \hat{\epsilon}_T(h, \hat{h}_T) + \frac{N_S}{N_S + N_T'} \left(\hat{\epsilon}_S(h, \hat{h}_S) + \sqrt{\frac{4d}{N_S}\log(\frac{eN_S}{d}) + \frac{1}{N_S}\log(\frac{1}{\delta})}\right.$$

$$\left. + 2C_f C_g W_1 \left(\mathbb{P}_S(G), \mathbb{P}_T(G)\right) + \omega'\right)$$

$$\leq \hat{\epsilon}_S(h, \hat{h}_S) + \sqrt{\frac{4d}{N_S}\log(\frac{eN_S}{d}) + \frac{1}{N_S}\log(\frac{1}{\delta})} + 2C_f C_g W_1(\mathbb{P}_S(G), \mathbb{P}_T(G)) + \omega,$$

*where $\omega = \min_{\|g\|_{Lip} \leq C_g, \|f\|_{Lip} \leq C_f} \{\epsilon_S(h, \hat{h}_S) + \epsilon_T(h, \hat{h}_S)\}$ and $\omega' = \min(|\epsilon_S(h, \hat{h}_S) - \epsilon_S(h, \hat{h}_T)|, |\epsilon_T(h, \hat{h}_S) - \epsilon_T(h - \hat{h}_T)|)$.*

The proof is detailed in Appendix A and B. Theorem 3 provides the theoretical guarantee of the iterative learning with Eq. 2 and 3. From Theorem 4, we observe that the bound of CoCA is lower than GDA by incorporating source and target samples during training, demonstrating that it is possible to design a more accurate model specifically tailored for graphs with a theoretical guarantee.

### 4.3 LEARNING FRAMEWORK

Through the iterative learning process of the MP and SP branches, we integrate them into a unified contrastive learning framework. Specifically, this framework employs cross-domain contrastive learning to achieve category-level domain consistency, while multi-view contrastive learning ensures consistent representation between the branches.

**Multi-view Contrastive Learning.** For each a graph $G_i$, we first obtain the embeddings from MP and SP branches, i.e., $\mathbf{z}_i^{MP}$ and $\mathbf{z}_i^{SP}$. Then, we introduce the InfoNCE loss to enhance the consistency representation cross coupled branches. Formally,

$$\mathcal{L}_{mv} = -\frac{1}{|\mathcal{D}^s| + |\mathcal{D}^t|} \sum_{G_i \in \mathcal{D}^s \cup \mathcal{D}^t} \log \frac{exp(\mathbf{z}_i^{MP} \cdot \mathbf{z}_i^{SP}/\tau)}{\sum_{G_j, j \neq i} exp(\mathbf{z}_i^{MP} \cdot \mathbf{z}_j^{SP}/\tau)},$$

where $\tau$ is the temperature parameter and set to $0.5$ as default (He et al., 2020).

**Cross-domain Contrastive Learning.** To achieve the category-level domain consistency, we construe the cross-domain contrastive learning with the help of pseudo-labels from each branch. Taking

Table 1: The classification results (in %) on Mutagenicity under edge density domain shift (source→target). M0, M1, M2, and M3 denote the sub-datasets partitioned with edge density. **Bold** results indicate the best performance.

| Methods | M0→M1 | M1→M0 | M0→M2 | M2→M0 | M0→M3 | M3→M0 | M1→M2 | M2→M1 | M1→M3 | M3→M1 | M2→M3 | M3→M2 | Avg. |
|---|---|---|---|---|---|---|---|---|---|---|---|---|---|
| WL subtree | 74.9 | 74.8 | 67.3 | 69.9 | 57.8 | 57.9 | 73.7 | 80.2 | 60.0 | 57.9 | 70.2 | 73.1 | 68.1 |
| GCN | $73.0_{\pm1.7}$ | $68.7_{\pm1.5}$ | $66.8_{\pm3.5}$ | $69.2_{\pm0.9}$ | $53.9_{\pm3.4}$ | $53.4_{\pm2.7}$ | $69.3_{\pm0.8}$ | $74.0_{\pm1.1}$ | $55.1_{\pm1.3}$ | $42.6_{\pm1.9}$ | $55.5_{\pm3.5}$ | $57.9_{\pm2.9}$ | 61.6 |
| GIN | $74.1_{\pm1.8}$ | $73.4_{\pm3.4}$ | $65.4_{\pm1.5}$ | $70.4_{\pm2.9}$ | $58.9_{\pm2.7}$ | $61.2_{\pm1.1}$ | $73.2_{\pm3.8}$ | $77.7_{\pm3.0}$ | $63.1_{\pm3.7}$ | $63.9_{\pm2.4}$ | $67.4_{\pm2.3}$ | $73.2_{\pm1.9}$ | 68.5 |
| GMT | $69.0_{\pm4.0}$ | $67.4_{\pm3.8}$ | $60.3_{\pm4.2}$ | $66.5_{\pm3.8}$ | $54.9_{\pm1.6}$ | $54.8_{\pm3.6}$ | $65.6_{\pm4.2}$ | $70.4_{\pm3.2}$ | $64.0_{\pm2.3}$ | $56.8_{\pm4.3}$ | $64.7_{\pm1.5}$ | $61.1_{\pm3.5}$ | 63.0 |
| CIN | $68.5_{\pm2.1}$ | $65.1_{\pm2.6}$ | $65.4_{\pm1.3}$ | $63.6_{\pm2.8}$ | $57.3_{\pm3.4}$ | $59.0_{\pm3.1}$ | $59.3_{\pm1.5}$ | $68.3_{\pm1.3}$ | $58.1_{\pm2.4}$ | $71.1_{\pm3.1}$ | $60.7_{\pm1.7}$ | $61.7_{\pm2.4}$ | 63.2 |
| CDAN | $74.2_{\pm0.3}$ | $73.7_{\pm0.5}$ | $68.8_{\pm0.2}$ | $71.8_{\pm0.4}$ | $59.9_{\pm2.0}$ | $58.6_{\pm1.9}$ | $70.7_{\pm1.4}$ | $74.3_{\pm0.3}$ | $59.2_{\pm1.2}$ | $69.0_{\pm1.6}$ | $60.0_{\pm1.2}$ | $62.7_{\pm1.3}$ | 66.9 |
| ToAlign | $75.5_{\pm1.9}$ | $67.1_{\pm3.8}$ | $68.1_{\pm1.5}$ | $63.3_{\pm2.7}$ | $55.6_{\pm1.2}$ | $67.3_{\pm4.3}$ | $69.4_{\pm3.3}$ | $77.0_{\pm1.2}$ | $57.6_{\pm1.6}$ | $74.9_{\pm2.4}$ | $59.0_{\pm3.3}$ | $64.6_{\pm3.4}$ | 66.6 |
| MetaAlign | $74.5_{\pm0.9}$ | $73.8_{\pm0.6}$ | $69.4_{\pm1.2}$ | $72.6_{\pm1.3}$ | $59.8_{\pm1.8}$ | $70.7_{\pm2.7}$ | $72.0_{\pm0.5}$ | $75.6_{\pm0.6}$ | $62.4_{\pm2.1}$ | $72.3_{\pm1.9}$ | $62.2_{\pm1.1}$ | $72.0_{\pm1.2}$ | 69.7 |
| DEAL | $76.3_{\pm0.2}$ | $72.4_{\pm0.7}$ | $68.8_{\pm1.0}$ | $72.5_{\pm0.7}$ | $57.6_{\pm0.6}$ | $67.6_{\pm1.9}$ | $77.4_{\pm0.6}$ | $80.0_{\pm0.7}$ | $64.9_{\pm0.7}$ | $72.8_{\pm1.4}$ | $70.3_{\pm0.3}$ | $76.2_{\pm1.3}$ | 71.4 |
| CoCo | $77.5_{\pm0.4}$ | $75.7_{\pm1.3}$ | $68.3_{\pm3.7}$ | $74.9_{\pm0.5}$ | $65.1_{\pm2.1}$ | $74.0_{\pm0.4}$ | $76.9_{\pm0.6}$ | $77.4_{\pm3.4}$ | $66.4_{\pm1.5}$ | $71.2_{\pm2.7}$ | $62.8_{\pm4.2}$ | $77.1_{\pm0.6}$ | 72.2 |
| SGDA | OOM | OOM | OOM | OOM | OOM | OOM | OOM | OOM | OOM | OOM | OOM | OOM | OOM |
| DGDA | OOM | OOM | OOM | OOM | OOM | OOM | OOM | OOM | OOM | OOM | OOM | OOM | OOM |
| A2GNN | $55.3_{\pm0.3}$ | $54.9_{\pm0.6}$ | $55.8_{\pm0.4}$ | $55.1_{\pm0.8}$ | $54.2_{\pm1.0}$ | $57.1_{\pm1.2}$ | $56.1_{\pm0.5}$ | $55.2_{\pm0.7}$ | $57.9_{\pm1.5}$ | $56.3_{\pm0.6}$ | $54.4_{\pm0.5}$ | $58.1_{\pm1.5}$ | 55.8 |
| PA-BOTH | $56.3_{\pm0.5}$ | $57.7_{\pm0.9}$ | $56.9_{\pm0.6}$ | $56.2_{\pm1.0}$ | $55.7_{\pm0.8}$ | $56.5_{\pm0.9}$ | $57.8_{\pm1.2}$ | $56.9_{\pm2.1}$ | $56.5_{\pm1.5}$ | $56.2_{\pm1.8}$ | $56.8_{\pm1.4}$ | $57.4_{\pm0.7}$ | 56.8 |
| CoCA | $\mathbf{82.4_{\pm1.5}}$ | $\mathbf{80.8_{\pm1.2}}$ | $\mathbf{74.5_{\pm1.7}}$ | $\mathbf{79.6_{\pm2.1}}$ | $\mathbf{74.8_{\pm2.2}}$ | $\mathbf{79.2_{\pm0.7}}$ | $\mathbf{83.4_{\pm0.9}}$ | $\mathbf{85.7_{\pm0.6}}$ | $\mathbf{73.9_{\pm0.8}}$ | $\mathbf{81.3_{\pm1.5}}$ | $\mathbf{77.8_{\pm0.7}}$ | $\mathbf{83.3_{\pm1.4}}$ | 79.7 |

the MP branch as an example, we first filter highly dependable samples with the threshold $\zeta$ as introduced in 4.2, and then calculate the loss between source and target domain with the same category.

$$\mathcal{L}_{cd} = - \sum_{j \in \Omega(j)} \frac{1}{|\Pi(j)|} \sum_{i \in \Pi(j)} \log \frac{exp\left(\mathbf{z}_j^{SP,t} \cdot \mathbf{z}_i^{SP,s}/\tau\right)}{\sum_{G_k \in \mathcal{D}^s, k \notin \Pi(i)} exp\left(\mathbf{z}_j^{SP,t} \cdot \mathbf{z}_k^{SP,s}/\tau\right)},$$

where $\Pi(j) = \{i|y_i^s = \hat{y}_j^t\}$ denotes the index of all positives in the source domain, $\Omega(i) = \{i|p_\theta(\hat{y}_i^t | G^s, G^t, Y^s) > \zeta\}$ is the index of filtered highly dependable samples from the target domain.

**Iterative Optimization.** As introduced in Section 4.2, we first filter the highly dependable samples in the MP branch, i.e., $\Omega(i) = \{i|p_\theta(\hat{y}_i^t | G^s, G^t, Y^s) > \zeta\}$, and then optimize the objective function to update $\phi$ in the SP branch:

$$\mathcal{L} = \mathcal{L}_1 + \alpha \mathcal{L}_{mv} + \beta \mathcal{L}_{cd}. \tag{5}$$

After that, we utilize the updated SP branch to filter the highly dependable samples, i.e., $\Omega(i) = \{i|q_\phi(\hat{y}_i^t | G^s, G^t, Y^s) > \zeta\}$, and update $\theta$ in the MP branch:

$$\mathcal{L} = \mathcal{L}_2 + \alpha \mathcal{L}_{mv} + \beta \mathcal{L}_{cd}, \tag{6}$$

where $\alpha$ and $\beta$ are the hyper-parameters. The complete algorithm for the proposed CoCA is summarized in Algorithm 1.

## 4.4 COMPLEXITY ANALYSIS.

The algorithmic complexity of CoCA consists of two main components: dual-branch learning and branch coupling. The complexity of the dual-branch learning process is given by $\mathcal{O}(LN(N + E + Kd) + LN^2d)$, where $L$ denotes number of layers in the network, $d$ is the dimension of features, $N$ is the number of nodes in the graph, $E$ is the number of edges, and $d$ is the maximum path length. The complexity of branch coupling, which involves the interaction between branches, is $\mathcal{O}(B^2d)$, where $B$ is the number of selected samples for coupling. Thus, the overall complexity of CoCA is $\mathcal{O}((LN^2 + B^2 + LNK)d + LNE)$.

CoCA's complexity is influenced by its hyperparameters $d$, $L$, $B$, and $K$. By appropriately controlling these values, the model's complexity can be managed effectively to ensure scalability. For small graphs, where $N \approx d$ and $d \approx E$, the complexity simplifies to $\mathcal{O}((LN^2 + B^2 + LNK)d)$. For large graphs, where $N \ll d$, $E \ll d$, $N \ll B$ with $K$ customarily set to a small value, the complexity reduces to $\mathcal{O}(LN^2d + LNE)$.

For large graphs, CoCA's complexity approximates that of the Graph Multiset Transformer (GMT), which is $\mathcal{O}(LN^2d)$. This demonstrates that CoCA maintains competitive scalability while incorporating additional capabilities for category-level alignment.

## 5 EXPERIMENTS

### 5.1 EXPERIMENTAL SETTINGS

**Datasets.** We use 4 graph classification benchmarks: Mutagenicity (M) (Kazius et al., 2005), FRANKENSTEIN (F) (Orsini et al., 2015), NCI1 (N) (Wale et al., 2008), and PROTEINS (P) (Dob-

Table 2: The graph classification results (in %) on FRANKENSTEIN under node domain shift (source→target). F0, F1, F2, and F3 denote the sub-datasets partitioned with node. **Bold** results indicate the best performance.

| Methods | F0→F1 | F1→F0 | F0→F2 | F2→F0 | F0→F3 | F3→F0 | F1→F2 | F2→F1 | F1→F3 | F3→F1 | F2→F3 | F3→F2 | Avg. |
|---|---|---|---|---|---|---|---|---|---|---|---|---|---|
| WL subtree | 65.7 | 71.8 | 57.9 | 71.1 | 47.4 | 43.4 | 65.5 | 75.1 | 45.3 | 34.9 | 52.7 | 49.8 | 56.7 |
| GCN | 70.6±2.1 | 60.3±1.5 | 60.5±3.4 | 62.3±1.1 | 58.4±0.5 | 43.2±0.2 | 63.8±1.0 | 70.3±0.3 | 50.6±1.0 | 32.8±0.3 | 50.1±0.4 | 42.2±0.2 | 55.4 |
| GIN | 66.7±2.1 | 73.7±2.4 | 57.3±3.1 | 69.4±2.3 | 58.6±0.4 | 43.1±0.3 | 66.4±2.7 | 74.8±1.8 | 42.2±1.6 | 33.5±1.0 | 57.4±0.8 | 43.9±2.3 | 57.2 |
| GMT | 67.3±0.3 | 56.8±0.4 | 58.0±0.2 | 56.8±0.2 | 60.6±0.3 | 56.8±0.5 | 57.8±0.1 | 67.3±0.1 | 39.5±0.3 | 67.3±0.2 | 39.5±0.5 | 57.8±0.4 | 57.1 |
| CIN | 67.6±0.4 | 63.7±2.1 | 58.9±1.0 | 56.8±0.4 | 63.6±0.4 | 59.5±2.7 | 58.7±1.2 | 67.0±0.5 | 61.7±1.6 | 67.8±0.7 | 62.2±2.1 | 56.0±1.3 | 61.9 |
| CDAN | 72.9±0.4 | 72.7±0.4 | 65.4±0.3 | 72.9±0.1 | 61.2±0.3 | 70.3±0.2 | 65.7±0.4 | 72.7±0.1 | 61.0±0.1 | 72.1±1.2 | 60.7±0.2 | 65.3±0.6 | 67.7 |
| ToAlign | 32.7±2.0 | 43.2±0.1 | 42.2±1.3 | 43.2±0.9 | 60.5±0.7 | 43.2±1.2 | 42.2±0.4 | 32.7±1.2 | 60.5±0.9 | 32.7±0.3 | 60.5±0.7 | 42.2±0.4 | 44.7 |
| MetaAlign | 67.3±0.7 | 56.8±0.2 | 57.8±0.6 | 56.8±0.4 | 60.5±1.3 | 56.8±0.8 | 57.8±0.1 | 67.3±1.1 | 60.5±0.4 | 67.3±0.6 | 60.5±0.7 | 57.8±0.6 | 60.6 |
| DEAL | 75.0±0.9 | 76.3±2.4 | 65.9±1.8 | 77.5±2.7 | 60.3±4.5 | 69.7±3.2 | 67.2±1.5 | 75.3±1.7 | 57.4±4.1 | 71.1±2.2 | 65.7±2.7 | 66.4±1.6 | 69.0 |
| CoCo | 74.2±1.7 | 74.3±0.6 | 65.9±1.2 | 72.7±2.1 | 61.1±0.2 | 71.0±1.7 | 68.6±0.3 | 75.9±0.2 | 60.7±0.2 | 73.9±0.4 | 59.7±1.1 | 67.3±0.8 | 68.8 |
| SGDA | 55.9±0.6 | 57.1±0.5 | 56.1±0.4 | 54.6±0.8 | 55.8±1.1 | 57.7±0.6 | 54.3±0.7 | 53.6±1.3 | 59.1±0.8 | 56.7±0.6 | 55.4±1.2 | 53.8±0.5 | 55.9 |
| DGDA | OOM | OOM | OOM | OOM | OOM | OOM | OOM | OOM | OOM | OOM | OOM | OOM | OOM |
| A2GNN | 55.9±0.7 | 55.7±0.4 | 56.6±0.6 | 57.1±1.0 | 56.1±1.2 | 55.8±0.5 | 56.5±0.7 | 55.5±0.4 | 55.9±0.8 | 56.2±0.6 | 56.5±1.5 | 56.0±0.5 | 56.2 |
| PA-BOTH | 56.4±0.5 | 55.9±0.6 | 56.0±0.5 | 56.4±0.4 | 56.3±0.6 | 57.7±0.7 | 56.6±0.2 | 58.8±0.9 | 56.9±0.7 | 57.2±0.3 | 56.5±0.5 | 58.3±0.8 | 56.9 |
| CoCA | **81.6±1.5** | **83.5±0.6** | **78.5±0.6** | **82.4±2.3** | **71.1±0.8** | **76.9±1.1** | **75.2±0.5** | **82.0±1.1** | **79.5±1.4** | **79.5±1.2** | **72.7±0.6** | **77.7±1.0** | **78.4** |

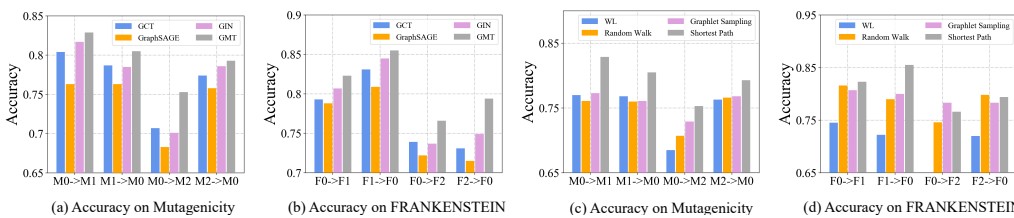

Figure 3: The performance with different GNNs and kernels on different datasets. (a), (b) are the performance of different GNNs, (c), (d) are the performance of different graph kernels.

son & Doig, 2003), obtained from TUDataset (Morris et al., 2020) to evaluate the effectiveness of the proposed CoCA. The details are presented in Appendix D. To assess the domain shift in each dataset, we follow (Yin et al., 2023) and partition each dataset into four sub-datasets ($D0$, $D1$, $D2$, and $D3$, where $D$ represents the respective dataset) based on edge and node density and graph flux.

**Baselines.** We compare the proposed CoCA with various state-of-the-art methods, including the kernel-based approach: WL subtree (Shervashidze et al., 2011), GNNs methods: GCN (Kipf & Welling, 2017a), GIN (Xu et al., 2019b), GMT (Baek et al., 2021), CIN (Bodnar et al., 2021), and recent domain adaptation methods: CDAN (Long et al., 2018), ToAlign (Wei et al., 2021b), MetaAlign (Wei et al., 2021a), and graph domain adaptation methods: DEAL (Yin et al., 2022), CoCo (Yin et al., 2023), SGDA (Qiao et al., 2023), DGDA (Cai et al., 2024), A2GNN (Liu et al., 2024a) and PA-BOTH (Liu et al., 2024b). The details are introduced in Appendix F and the implementation details are proposed in Appendix E.

## 5.2 PERFORMANCE COMPARISON

Table 1, 2 and 3 show the comparison performance of CoCA and baselines on Mutagenicity, FRANKENSTEIN and NCI1 datasets under different domain shift. More results are shown in Appendix G. From the results, we find that: (1) The GDA methods, including DEAL, CoCo, CoCA, etc., consistently outperform the kernel and GNN methods. This demonstrates that domain shift limits the expressive capability of traditional graph methods. Therefore, it is critical to design the domain invariant methods for GDA. (2) The GDA methods demonstrate competitive performance compared to traditional domain adaptation approaches. This achievement can be attributed to the challenges associated with obtaining high-quality graph representations, which make the direct application of domain adaptation techniques to graphs a demanding task. (3) The proposed CoCA outperforms recent GDA methods. We attribute this performance gain to two factors: (i) The dual branch approach for graph semantic extraction, which effectively leverages the complementary strengths of message passing and shortest path aggregation models. (ii) The architecture of the branch coupling module effectively aligns the category-level distribution, addressing the category-agnostic limitations typically encountered with traditional GDA methods.

Table 3: The graph classification results (in %) on NCI1 under graph flux domain shift (source→target). N0, N1, N2, and N3 denote the sub-datasets partitioned with graph flux. **Bold** results indicate the best performance. OOM means out of memory.

| Methods | N0→N1 | N1→N0 | N0→N2 | N2→N0 | N0→N3 | N3→N0 | N1→N2 | N2→N1 | N1→N3 | N3→N1 | N2→N3 | N3→N2 | Avg. |
|---|---|---|---|---|---|---|---|---|---|---|---|---|---|
| WL subtree | 75.9 | 70.4 | 64.3 | 63.9 | 60.6 | 64.7 | 73.2 | **78.9** | 66.8 | 69.2 | 74.2 | 72.9 | 69.6 |
| GCN | 49.2±1.7 | 55.8±1.5 | 46.8±0.5 | 54.6±2.2 | 43.4±0.6 | 46.7±0.2 | 50.0±1.8 | 57.2±2.2 | 44.2±0.4 | 51.6±0.8 | 62.7±2.1 | 56.8±1.3 | 51.6 |
| GIN | 68.8±2.5 | 70.6±1.0 | 64.2±1.1 | 67.2±2.4 | 62.2±1.8 | 62.5±1.5 | 68.7±2.4 | 72.5±0.6 | 63.3±1.6 | 65.2±0.6 | 62.4±0.3 | 70.9±0.5 | 66.6 |
| GMT | 66.7±0.3 | 58.2±0.5 | 63.9±0.3 | 58.4±0.3 | 63.8±0.4 | 56.7±0.5 | 63.9±0.7 | 66.3±1.0 | 63.8±1.1 | 66.6±0.4 | 63.8±0.2 | 62.6±0.7 | 62.9 |
| CIN | 58.7±2.4 | 54.9±0.2 | 52.0±0.3 | 54.8±0.1 | 56.6±0.2 | 54.9±0.1 | 52.9±1.4 | 52.8±0.5 | 56.5±0.6 | 52.8±2.1 | 58.5±0.8 | 56.6±1.4 | 55.1 |
| CDAN | 64.0±1.1 | 68.1±0.3 | 60.1±0.5 | 64.0±1.3 | 60.9±0.2 | 57.8±1.0 | 64.3±1.6 | 61.2±0.2 | 66.3±0.7 | 59.0±0.5 | 68.9±0.3 | 63.7±0.6 | 63.2 |
| ToAlign | 52.8±0.5 | 54.8±0.2 | 48.2±1.1 | 54.8±1.5 | 44.0±0.8 | 54.8±2.0 | 48.2±1.7 | 52.8±0.6 | 44.0±0.2 | 52.8±0.3 | 44.0±1.0 | 48.2±1.2 | 50.0 |
| MetaAlign | 63.1±0.3 | 63.8±1.3 | 58.9±2.4 | 58.5±0.4 | 59.1±2.1 | 59.2±1.6 | 70.1±0.8 | 63.3±1.4 | 66.5±2.7 | 60.9±1.1 | 71.4±0.2 | 67.5±0.8 | 63.5 |
| DEAL | 70.7±0.9 | 72.3±0.2 | 69.9±0.8 | 68.9±0.7 | 64.1±0.6 | 65.6±0.9 | 71.9±0.4 | 69.9±1.7 | 70.6±0.4 | 66.5±0.3 | 71.6±0.7 | 69.9±0.5 | 69.3 |
| CoCo | 64.0±1.3 | 63.9±0.6 | 65.8±1.8 | 59.9±1.7 | 62.2±2.1 | 60.6±1.6 | 65.0±2.1 | 64.8±1.4 | 60.0±0.8 | 61.3±0.5 | 68.5±0.4 | 67.1±0.6 | 63.6 |
| SGDA | OOM | OOM | OOM | OOM | OOM | OOM | OOM | OOM | OOM | OOM | OOM | OOM | OOM |
| DGDA | OOM | OOM | OOM | OOM | OOM | OOM | OOM | OOM | OOM | OOM | OOM | OOM | OOM |
| A2GNN | 56.5±0.9 | 56.7±0.7 | 58.8±1.2 | 56.0±1.0 | 61.2±1.5 | 60.9±1.6 | 61.0±1.3 | 56.1±1.9 | 64.9±1.6 | 59.3±2.1 | 65.4±1.5 | 63.3±2.3 | 60.1 |
| PA-BOTH | 57.4±0.5 | 58.2±0.4 | 58.2±0.6 | 57.6±0.8 | 58.2±0.6 | 58.5±0.5 | 58.1±1.0 | 59.9±0.7 | 63.6±1.1 | 57.7±0.9 | 58.2±0.8 | 57.6±1.2 | 58.7 |
| CoCA | **81.4±0.9** | **76.3±1.5** | **75.1±0.7** | **74.3±1.2** | **72.6±1.5** | **78.4±0.8** | **77.4±2.3** | 73.2±2.0 | **75.3±0.5** | **76.9±1.0** | **80.1±1.3** | **74.0±0.3** | **76.3** |

Table 4: The results of ablation studies on Mutagenicity (source→target). **Bold** results indicate the best performance.

| Methods | M0→M1 | M1→M0 | M0→M2 | M2→M0 | M0→M3 | M3→M0 | M1→M2 | M2→M1 | M1→M3 | M3→M1 | M2→M3 | M3→M2 | Avg. |
|---|---|---|---|---|---|---|---|---|---|---|---|---|---|
| CoCA-MP | 77.3 | 70.1 | 70.8 | 71.6 | 68.3 | 71.1 | 77.2 | 82.8 | 68.3 | 77.6 | 67.8 | 76.2 | 73.2 |
| CoCA-SP | 80.6 | 74.1 | 68.8 | 70.8 | 65.7 | 72.5 | 78.3 | 83.3 | 67.2 | 78.5 | 69.6 | 81.3 | 74.2 |
| CoCA/BC | 76.0 | 76.3 | 69.3 | 74.4 | 67.4 | 64.9 | 78.8 | 82.2 | 68.5 | 73.2 | 69.4 | 77.5 | 73.2 |
| CoCA/MV | 77.3 | 77.2 | 71.8 | 76.4 | 70.2 | 75.7 | 79.3 | 83.1 | 71.4 | 79.5 | 74.2 | 78.5 | 76.2 |
| CoCA/CD | 78.1 | 75.4 | 70.4 | 76.7 | 72.7 | 77.2 | 78.8 | **86.4** | 72.3 | 80.1 | 73.4 | 79.3 | 76.7 |
| CoCA | **82.9** | **80.5** | **75.3** | **79.3** | **74.4** | **79.2** | **83.1** | 86.1 | **74.7** | **81.3** | **78.5** | **82.6** | **79.8** |

## 5.3 FLEXIBILITY OF CoCA

To show the flexibility of CoCA, we replace the MP and SP branches with different GNNs and kernels. Specifically, we replace the MP branch with GCN (Kipf & Welling, 2017a), GIN (Xu et al., 2019a) and Graphsage (Hamilton et al., 2017), and the SP branch with Graph Sampling (Leskovec & Faloutsos, 2006), Random Walk (Kalofolias et al., 2021) and WL kernel (Neumann et al., 2016). Figure 3 shows the performance of different GNNs and graph kernels on four datasets, and we have similar observations on other datasets. More results are shown in Appendix H. From the results, we observe that when compared to other GNNs and graph kernels, GMT and shortest path aggregation consistently achieve the best performance in most cases. This can be attributed to the powerful representation capabilities of GMT and the shortest path kernel. This observation further justifies our choice of GMT and shortest path aggregation to improve performance in our GDA task.

## 5.4 ABLATION STUDY

To assess the impact of each module on CoCA, we conduct ablation experiments with various configurations: (1) CoCA-MP, where both branches exclusively use the message passing model; (2) CoCA-SP, where both branches exclusively use the path aggregation model; (3) CoCA/BC, removal of the branch coupling module; (4) CoCA/MV, removal of the multi-view contrastive learning module; (5) CoCA/CD, removal of the cross-domain contrastive learning module.

We conducted these experiments on the Mutagenicity dataset, and the results are presented in Table 4. From the results, we observe that: (1) CoCA outperforms both CoCA-MP and CoCA-SP, underscoring the importance of extracting graph semantics from both implicit and explicit perspectives. (2) The performance of CoCA is significantly superior to models without the branch coupling module (i.e., CoCA/BC), indicating that by aligning the category-level distribution, the branch coupling module effectively addresses the category-agnostic issue caused by aligning the entire feature distribution. (3) CoCA/MV and CoCA/CD perform worse than CoCA. We attribute this to the fact that by disregarding multi-view and cross-domain contrastive learning, CoCA cannot efficiently learn consistent representations between the source and target, leading to diminished predictive performance.

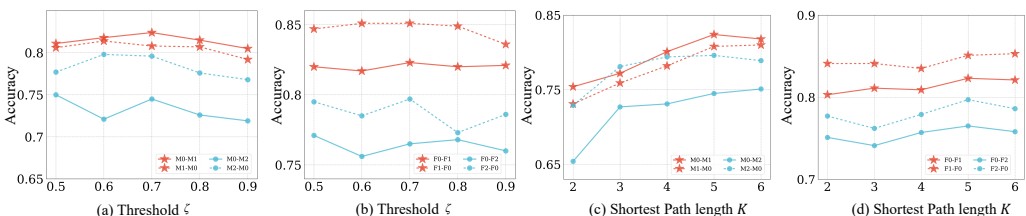

Figure 4: Hyperparameter sensitivity of threshold $\zeta$ and shortest path length $K$ on different datasets. (a), (b) are the performance of threshold $\zeta$, (c), (d) are the performance of shortest path length $K$.

## 5.5 SENSITIVITY ANALYSIS

In this part, we investigate the influence of hyperparameters on the performance of the proposed CoCA. We specifically examine the effects of two key hyperparameters, including the threshold $\zeta$ in the branch coupling module for category alignment, and the shortest path length $K$ in the SP branch. We report the results of $\zeta$ and $K$ in Figure 4. $\zeta$ determines the number of reliable samples selected from each branch, and we vary $\zeta$ in the range from 0.5 to 0.9. The experimental results presented in Figure 4 (a), (b) indicate an initial increase followed by stability or a decreasing trend in performance as $\zeta$ increases. We attribute the reason to the fact that smaller values of $\zeta$ introduce low-confidence samples, which would detriment the performance of CoCA. Conversely, larger values of $\zeta$ introduce high-confidence samples for training. However, excessively high values of $\zeta$ may lead to fewer filtered samples, potentially resulting in a decline in model performance. Therefore, we set $\zeta$ to 0.7 as default. Additionally, the parameter $K$ controls the number of shortest paths extracted in the SP branch, and we vary $K$ in the range of $\{2, 3, 4, 5, 6\}$. The results are shown in Figure 4 (c) and (d). From the results, we observe that increasing $K$ generally leads to improved performance when the value is small. This suggests that incorporating more shortest path aggregations can enhance the representation capability. However, when $K$ becomes large, the performance stays stable. Considering the significant increase in algorithmic complexity associated with higher values of $K$, we set $K = 5$. Additionally, we examine the accuracy of filtered samples from MP and SP branch within the branch coupling module, and the results are shown in Appendix I.

## 6 CONCLUSION

In this paper, we study a practical problem of unsupervised graph classification and propose a novel approach named CoCA. We utilize the dual branch, i.e., a message passing and a shortest path aggregation branch, to explore the graph semantics from the implicit and explicit perspectives. The introduction of the branch coupling module ensures effective category-level alignment, mitigating the category-agnostic issues typically encountered in traditional GDA methods. Furthermore, the framework incorporates cross-domain and multi-view contrastive learning, enhancing the consistency of representations across domains and branches. Theoretical analysis demonstrates that CoCA achieves a tighter empirical risk bound compared to existing GDA methods. Extensive experiments across multiple datasets validate the superiority of CoCA in handling domain shifts. In future work, we aim to explore extending CoCA to more complex scenarios, such as domain generalization and source-free domain adaptation.

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

# A APPENDIX

## A PROOF OF THEOREM 3

**Theorem 3** *The iterative learning process (i.e., Eq. 2 and 3) follows the optimization of maximizing the Evidence Lower Bound (ELBO):*

$$\log p_\theta\left(Y^s \mid G^s, G^t\right) \geq \mathbb{E}_{q_\phi(Y^t|G^t)}\left[\log p_\theta\left(Y^s, Y^t \mid G^s, G^t\right) - \log q_\phi\left(Y^t \mid G^t\right)\right]. \quad (7)$$

*Proof.* Considering the message passing branch (MP branch) and the shortest path aggregation branch (SP branch), our objective is to identify highly dependable pseudo-labels in the target domain and integrate them with the source domain to fine-tune the model. In this way, we can efficiently align the category distribution. Nevertheless, with the challenges of category discrepancy and error accumulation, we cannot achieve satisfactory pseudo-labels in a signal branch. To address the issue, we introduce the distribution $p_\theta(\cdot)$ and $q_\phi(\cdot)$ for MP and SP branch, and aim to maximize the evidence lower bound (ELBO) of the log-likelihood with the source and target labels $Y^s$ and $Y^t$:

$$\log p_\theta\left(Y^s \mid G^s, G^t\right) \geq \mathbb{E}_{q_\phi(Y^t|G^t)}\left[\log p_\theta\left(Y^s, Y^t \mid G^s, G^t\right) - \log q_\phi\left(Y^t \mid G^t\right)\right]. \quad (8)$$

To maximize Eq. 8, we alternatively optimize the SP branch with distribution $q_\phi\left(Y^t \mid G^t\right)$ and the MP branch with $p_\theta\left(Y^s, Y^t \mid G^s, G^t\right)$. Firstly, we fix the MP branch and update the SP branch by minimizing $\mathbb{E}_{q_\phi(Y^t|G^t)} \log q_\phi\left(Y^t \mid G^t\right)$. To avoid error accumulation, we take the help of MP branch to provide additional target graph information for SP branch optimization. In formulation, we calculate the negative KL divergence between $q_\phi\left(Y^t \mid G^t\right)$ and $p_\theta\left(Y^t \mid G^s, G^t, Y^s\right)$:

$$-KL\left(q_\phi\left(Y^t \mid G^t\right) \| p_\theta\left(Y^t \mid G^s, G^t, Y^s\right)\right) = \mathbb{E}_{p_\theta(Y^t|G^s, G^t, Y^s)}[\log q_\phi\left(Y^t \mid G^t\right)] + Cons,$$

where $Cons$ is a constant. To avoid the category shift, we further select the highly dependable pseudo-labels from the MP branch with a threshold $\zeta$, and the objective is:

$$\mathcal{L}_1 = \mathbb{E}_{p_\theta(\hat{y}_i^t|G^s, G^t, Y^s)>\zeta}\left[\log q_\phi\left(\hat{y}_i^t \mid G_i^t\right)\right] + \sum_{i=1}^{N_s} \log q_\phi\left(Y_i^s \mid G_i^s\right), \quad (9)$$

where $\hat{y}^t$ is the target graph pseudo-labels filtered from the MP branch. Similarly, we utilize the target samples filtered from the SP branch to support the training of the MP branch:

$$\mathcal{L}_2 = \mathbb{E}_{q_\phi(\hat{y}_i^t|G^s, G^t, Y^s)>\zeta}\left[\log p_\theta\left(\hat{y}_i^t \mid G_i^t\right)\right] + \sum_{i=1}^{N_s} \log p_\theta(Y_i^s \mid G_i^s). \quad (10)$$

The interactive optimization of the MP and SP branches offers two advantages. First, by incorporating highly confident target pseudo-labels into source domain training, we can effectively align the category distribution. Second, the pseudo-labels filtered from the other branch help mitigate the error accumulation issue caused by the single model.

## B PROOF OF THEOREM 4

**Theorem 4** *Under the assumption of Theorem 2, we further assume that there exists a small amount of i.i.d. samples with pseudo labels $\{(G_n, Y_n)\}_{n=1}^{N_T'}$ from the target distribution $\mathbb{P}_T(G, Y)$ $(N_T' \ll N_S)$ and bring in the conditional shift assumption that domains have different labeling function $\hat{h}_S \neq \hat{h}_T$ and $\max_{G_1, G_2} \frac{|\hat{h}_D(G_1) - \hat{h}_D(G_2)|}{\eta(G_1, G_2)} = C_h \leq C_f C_g (D \in \{S, T\})$ for some constant $C_h$ and distance measure $\eta$. Let $\mathcal{H} := \{h : \mathcal{G} \to \mathcal{Y}\}$ be the set of bounded real-valued functions with the pseudo-dimension $Pdim(\mathcal{H}) = d$, with probability at least $1 - \delta$ the following inequality holds:*

$$\epsilon_T(h, \hat{h}_T) \leq \frac{N_T'}{N_S + N_T'}\hat{\epsilon}_T(h, \hat{h}_T) + \frac{N_S}{N_S + N_T'}\left(\hat{\epsilon}_S(h, \hat{h}_S) + \sqrt{\frac{4d}{N_S}\log(\frac{eN_S}{d}) + \frac{1}{N_S}\log(\frac{1}{\delta})}\right.$$

$$\left. + 2C_f C_g W_1\left(\mathbb{P}_S(G), \mathbb{P}_T(G)\right) + \omega\right)$$

$$\leq \hat{\epsilon}_S(h, \hat{h}) + \sqrt{\frac{4d}{N_S}\log(\frac{eN_S}{d}) + \frac{1}{N_S}\log(\frac{1}{\delta})} + 2C_f C_g W_1(\mathbb{P}_S(G), \mathbb{P}_T(G)) + \omega,$$

$$(11)$$

*where* $\omega = \min_{||g||_{Lip} \leq C_g, ||f||_{Lip} \leq C_f} \{\epsilon_S(h, \hat{h}_S) + \epsilon_T(h, \hat{h}_S)\}$.

*Proof.* Before showing the designated lemma, we first introduce the following inequality to be used that:

$$
\begin{aligned}
|\epsilon_S(h, \hat{h}_S) - \epsilon_T(h, \hat{h}_T)| &= |\epsilon_S(h, \hat{h}_S) - \epsilon_S(h, \hat{h}_T) + \epsilon_S(h, \hat{h}_T) - \epsilon_T(h, \hat{h}_T)| \\
&\leq |\epsilon_S(h, \hat{h}_S) - \epsilon_S(h, \hat{h}_T)| + |\epsilon_S(h, \hat{h}_T) - \epsilon_T(h, \hat{h}_T)| \\
&\overset{(a)}{\leq} |\epsilon_S(h, \hat{h}_S) - \epsilon_S(h, \hat{h}_T)| + 2C_f C_g W_1\left(\mathbb{P}_S(G), \mathbb{P}_T(G)\right),
\end{aligned}
\tag{12}
$$

where $(a)$ results from (Shen et al., 2018) Theorem 1 with the assumption $\max(||h||_{Lip}, \max_{G_1, G_2} \frac{|\hat{h}_D(G_1) - \hat{h}_D(G_2)|}{\eta(G_1, G_2)}) \leq C_f C_g, D \in \{S, T\}$. Similarly, we obtain:

$$
|\epsilon_S(h, \hat{h}_S) - \epsilon_T(h, \hat{h}_T)| \leq |\epsilon_T(h, \hat{h}_S) - \epsilon_T(h, \hat{h}_T)| + 2C_f C_g W_1(\mathbb{P}_S(G), \mathbb{P}_T(G)).
\tag{13}
$$

We therefore combine them into:

$$
\begin{aligned}
|\epsilon_S(h, \hat{h}_S) - \epsilon_T(h, \hat{h}_T)| \leq &\,2C_f C_g W_1(\mathbb{P}_S(G), \mathbb{P}_T(G)) \\
&+ \min\left(|\epsilon_S(h, \hat{h}_S) - \epsilon_S(h, \hat{h}_T)|, |\epsilon_T(h, \hat{h}_S) - \epsilon_T(h, \hat{h}_T)|\right),
\end{aligned}
\tag{14}
$$

i.e. the following holds to bound the target risk $\epsilon_T(h, \hat{h}_T)$:

$$
\begin{aligned}
\epsilon_T(h, \hat{h}_T) \leq &\,\epsilon_S(h, \hat{h}_S) + 2C_f C_g W_1\left(\mathbb{P}_S(G), \mathbb{P}_T(G)\right) \\
&+ \min\left(|\epsilon_S(h, \hat{h}_S) - \epsilon_S(h, \hat{h}_T)|, |\epsilon_T(h, \hat{h}_S) - \epsilon_T(h, \hat{h}_T)|\right).
\end{aligned}
\tag{15}
$$

We next link the bound with the empirical risk and labeled sample size by showing, with probability at least $1 - \delta$ that:

$$
\begin{aligned}
\epsilon_T(h, \hat{h}_T) \leq &\,\epsilon_S(h, \hat{h}_S) + 2C_f C_g W_1\left(\mathbb{P}_S(G), \mathbb{P}_T(G)\right) \\
&+ \min\left(|\epsilon_S(h, \hat{h}_S) - \epsilon_S(h, \hat{h}_T)|, |\epsilon_T(h, \hat{h}_S) - \epsilon_T(h, \hat{h}_T)|\right) \\
\leq &\,\hat{\epsilon}_S(h, \hat{h}_S) + 2C_f C_g W_1\left(\mathbb{P}_S(G), \mathbb{P}_T(G)\right) \\
&+ \min\left(|\epsilon_S(h, \hat{h}_S) - \epsilon_S(h, \hat{h}_T)|, |\epsilon_T(h, \hat{h}_S) - \epsilon_T(h, \hat{h}_T)|\right) \\
&+ \sqrt{\frac{2d}{N_S} \log(\frac{eN_S}{d})} + \sqrt{\frac{1}{2N_S} \log(\frac{1}{\delta})},
\end{aligned}
\tag{16}
$$

and:

$$
\epsilon_T(h, \hat{h}_T) \leq \hat{\epsilon}_T(h, \hat{h}_T) + \sqrt{\frac{2d}{N'_T} \log(\frac{eN'_T}{d})} + \sqrt{\frac{1}{2N'_T} \log(\frac{1}{\delta})},
\tag{17}
$$

which results from (Mohri et al., 2018) Theorem 11.8. Lastly, we combine the above two inequalities, with probability at least $1 - \delta$ that:

$$
\begin{aligned}
\epsilon_T(h, \hat{h}_T) \overset{(a)}{\leq} &\,\frac{N'_T}{N_S + N'_T} \left(\hat{\epsilon}_T(h, \hat{h}_T) + \sqrt{\frac{2d}{N'_T} \log(\frac{eN'_T}{d})} + \sqrt{\frac{1}{2N'_T} \log(\frac{1}{\delta})}\right) \\
&+ \frac{N_S}{N_S + N'_T} \left(\hat{\epsilon}_S(h, \hat{h}_S) + \sqrt{\frac{2d}{N_S} \log(\frac{eN_S}{d})} + \sqrt{\frac{1}{2N_S} \log(\frac{1}{\delta})}\right) \\
&+ \frac{N_S}{N_S + N'_T} \Bigg(2C_f C_g W_1\left(\mathbb{P}_S(G), \mathbb{P}_T(G)\right) \\
&+ \min\left(|\epsilon_S(h, \hat{h}_S) - \epsilon_S(h, \hat{h}_T)|, |\epsilon_T(h, \hat{h}_S) - \epsilon_T(h, \hat{h}_T)|\right)\Bigg) \\
\overset{(b)}{\leq} &\,\frac{N'_T}{N_S + N'_T} \left(\hat{\epsilon}_T(h, \hat{h}_T) + \sqrt{\frac{4d}{N'_T} \log(\frac{eN'_T}{d})} + \frac{1}{N'_T} \log(\frac{1}{\delta})\right)
\end{aligned}
$$

$$+ \frac{N_S}{N_S + N_T'} \left( \hat{\epsilon}_S(h, \hat{h}_S) + \sqrt{\frac{4d}{N_S} \log(\frac{eN_S}{d}) + \frac{1}{N_S} \log(\frac{1}{\delta})} \right)$$

$$+ \frac{N_S}{N_S + N_T'} \left( 2C_f C_g W_1 \left( \mathbb{P}_S(G), \mathbb{P}_T(G) \right) \right.$$

$$+ \min \left( |\epsilon_S(h, \hat{h}_S) - \epsilon_S(h, \hat{h}_T)|, |\epsilon_T(h, \hat{h}_S) - \epsilon_T(h, \hat{h}_T)| \right) \right)$$

$$\overset{(c)}{\leq} \frac{N_T'}{N_S + N_T'} \hat{\epsilon}_T(h, \hat{h}_T) + \frac{N_S}{N_S + N_T'} \hat{\epsilon}_S(h, \hat{h}_S)$$

$$+ \frac{N_S}{N_S + N_T'} \left( 2C_f C_g W_1 \left( \mathbb{P}_S(G), \mathbb{P}_T(G) \right) \right.$$

$$+ \min \left( |\epsilon_S(h, \hat{h}_S) - \epsilon_S(h, \hat{h}_T)|, |\epsilon_T(h, \hat{h}_S) - \epsilon_T(h, \hat{h}_T)| \right) \right)$$

$$+ \frac{N_T'}{N_S + N_T'} \sqrt{\frac{4d}{N_T'} \log(\frac{eN_T'}{d}) + \frac{1}{N_T'} \log(\frac{1}{\delta})} + \frac{N_S}{N_S + N_T'} \sqrt{\frac{4d}{N_S} \log(\frac{eN_S}{d}) + \frac{1}{N_S} \log(\frac{1}{\delta})}$$

$$\doteq \frac{N_T'}{N_S + N_T'} \hat{\epsilon}_T(h, \hat{h}_T) + \frac{N_S}{N_S + N_T'} \hat{\epsilon}_S(h, \hat{h}_S) + \frac{N_S}{N_S + N_T'} \sqrt{\frac{4d}{N_S} \log(\frac{eN_S}{d}) + \frac{1}{N_S} \log(\frac{1}{\delta})}$$

$$+ \frac{N_S}{N_S + N_T'} \left( 2C_f C_g W_1 \left( \mathbb{P}_S(G), \mathbb{P}_T(G) \right) \right.$$

$$+ \min \left( |\epsilon_S(h, \hat{h}_S) - \epsilon_S(h, \hat{h}_T)|, |\epsilon_T(h, \hat{h}_S) - \epsilon_T(h, \hat{h}_T)| \right) \right)$$

$$= \frac{N_T'}{N_S + N_T'} \hat{\epsilon}_T(h, \hat{h}_T) + \frac{N_S}{N_S + N_T'} \left( \hat{\epsilon}_S(h, \hat{h}_S) + \sqrt{\frac{4d}{N_S} \log(\frac{eN_S}{d}) + \frac{1}{N_S} \log(\frac{1}{\delta})} \right.$$

$$+ 2C_f C_g W_1 \left( \mathbb{P}_S(G), \mathbb{P}_T(G) \right)$$

$$+ \min \left( |\epsilon_S(h, \hat{h}_S) - \epsilon_S(h, \hat{h}_T)|, |\epsilon_T(h, \hat{h}_S) - \epsilon_T(h, \hat{h}_T)| \right) \right)$$

where (a) is the outcome of applying the union bound with coefficient $\frac{N_T'}{N_S + N_T'}$, $\frac{N_S}{N_S + N_T'}$ respectively; (b) and (c) result from the Cauchy-Schwartz inequality and (c) additionally adopt the assumption $N_T' \ll N_S$, following the sleight-of-hand in (Li et al., 2021) Theorem 3.2.

Due to the sampels are selected with high confidence, thus, we have the following assumption:

$$\hat{\epsilon}_T \leq \epsilon_T \leq \hat{\epsilon}_S(h, \hat{h}) + \sqrt{\frac{4d}{N_S} \log(\frac{eN_S}{d}) + \frac{1}{N_S} \log(\frac{1}{\delta})} + 2C_f C_g W_1(\mathbb{P}_S(G), \mathbb{P}_T(G)) + \omega', \quad (18)$$

where $\omega' = \min_{||g||_{Lip} \leq C_g, ||f||_{Lip} \leq C_f} \{ \epsilon_S(h, \hat{h}) + \epsilon_T(h, \hat{h}) \}$, $\hat{\epsilon}_T$ is the empirical risk on the high confidence samples, $\epsilon_T$ is the empirical risk on the target domain. Besides, we have:

$$\min(|\epsilon_S(h, \hat{h}_S) - \epsilon_S(h, \hat{h}_T)|, |\epsilon_T(h, \hat{h}_S) - \epsilon_T(h, \hat{h}_T)|) \leq \min(\epsilon_S(h, \hat{h}_S) + \epsilon_T(h, \hat{h}_S)) \quad (19)$$

Then,

$$\epsilon_T(h, \hat{h}_T) \leq \frac{N_T'}{N_S + N_T'} \hat{\epsilon}_T(h, \hat{h}_T) + \frac{N_S}{N_S + N_T'} \left( \hat{\epsilon}_S(h, \hat{h}_S) + \sqrt{\frac{4d}{N_S} \log(\frac{eN_S}{d}) + \frac{1}{N_S} \log(\frac{1}{\delta})} \right.$$

$$+ 2C_f C_g W_1 \left( \mathbb{P}_S(G), \mathbb{P}_T(G) \right) + \omega \right)$$

$$\leq \hat{\epsilon}_S(h, \hat{h}) + \sqrt{\frac{4d}{N_S} \log(\frac{eN_S}{d}) + \frac{1}{N_S} \log(\frac{1}{\delta})} + 2C_f C_g W_1(\mathbb{P}_S(G), \mathbb{P}_T(G)) + \omega'.$$

$$(20)$$

## C  ALGORITHM

---

**Algorithm 1** Learning Algorithm of CoCA

---

**Input:** Source data $\mathcal{D}^s$; Target data $\mathcal{D}^t$.
**Output**: Parameters $\theta$ and $\phi$ for two branches.

---

 1: // Dual Graph Branch for Semantics Mining
 2: Initialize $\theta$ and $\phi$.
 3: Warm up the SP and MP branch to update $\theta$ and $\phi$.
 4: **while** not convergence **do**
 5:     // Branch Coupling for Category Alignment
 6:     Filter target pseudo-labels with the MP branch;
 7:     Optimize parameters $\phi$ with fixed $\theta$ by Eq. 5;
 8:     Filter target pseudo-labels with the SP branch;
 9:     Optimize parameters $\theta$ with fixed $\phi$ by Eq. 6;
10: **end while**

---

## D  INTRODUCTION OF DATASETS

We briefly introduce the datasets as follows:

- **Mutagenicity** (Wale et al., 2008): Mutagenicity includes 4,337 molecular structures and their corresponding Ames test results, comprising 2,401 mutagens and 1,936 non-mutagens, each represented as a graph.

- **FRANKENSTEIN** (Orsini et al., 2015): FRANKENSTEIN is a composite dataset created by merging the BURSI and MNIST datasets. Each data point is represented as a graph, with vertices corresponding to chemical atom symbols and edges indicating bond types.

- **PROTEINS** (Wale et al., 2008): PROTEINS is a dataset of protein graphs, where nodes represent amino acids, and edges signify connections between amino acids that are within a distance of less than 6 Angstroms.

- **NCI1** (Dobson & Doig, 2003): NCI1 consists of chemical molecules and compounds, using atoms as nodes and bonds as edges. With a total of 4,100 compounds, labels are assigned to determine whether a compound exhibits characteristics inhibiting cancer cell growth.

## E  IMPLEMENTATION DETAILS

In our CoCA, we employ GMT (Baek et al., 2021) in the MP branch and the shortest path model (Abboud et al., 2022) in the SP branch. For the SP branch, the maximum path lengths $K$ are set to 5 for all the datasets. For the adaptive perturbation, we set the $T = 5$ for perturbation learning. We warm up the dual branch for 50 epochs and update the branch coupling module 10 times. The pseudo-label filtering threshold $\zeta$ for the target dataset is set to 0.7. For all the methods, we use one of the sub-datasets as source data and the remaining as the target data for performance comparison. We set the hidden size to 128 and the learning rate to 0.001 as default. All the experiments are conducted on the same device, equipped with NVIDIA A6000 GPU.

## F  INTRODUCTION OF BASELINES

We introduce the baselines as follows:

- **WL subtree** (Shervashidze et al., 2011): The WL subtree functions as a kernel technique, gauging the resemblance among graphs through the designated kernel function.

- **GCN** (Kipf & Welling, 2017a): The fundamental concept of GCN is to update each central node by incorporating neighborhood information, resulting in an iterative generation of a representation vector.

- **GIN** (Xu et al., 2019b): GIN is a widely recognized neural network employing message passing, known for its enhanced expressive capabilities.

Table 5: Graph classification accuracy (in %) on FRANKENSTEIN under edge density domain shift (source→target). F0, F1, F2, and F3 denote the sub-datasets partitioned with edge density. **Bold** results indicate the best performance.

| Methods | F0→F1 | F1→F0 | F0→F2 | F2→F0 | F0→F3 | F3→F0 | F1→F2 | F2→F1 | F1→F3 | F3→F1 | F2→F3 | F3→F2 | Avg. |
|---|---|---|---|---|---|---|---|---|---|---|---|---|---|
| WL subtree | 71.6 | 72.1 | 62.1 | 71.2 | 57.8 | 67.7 | 64.0 | 75.3 | 41.1 | 59.2 | 55.9 | 55.4 | 62.8 |
| GCN | 66.5±0.4 | 60.0±0.8 | 55.4±0.3 | 60.0±0.1 | 39.6±0.3 | 40.0±0.4 | 55.4±0.2 | 66.5±0.1 | 39.6±0.6 | 33.5±0.3 | 39.6±0.1 | 44.7±0.2 | 50.1 |
| GIN | 71.4±4.7 | 73.4±3.4 | 60.8±2.7 | 66.0±3.4 | 50.5±3.7 | 51.6±1.8 | 64.8±1.0 | 71.3±3.5 | 48.3±4.2 | 57.4±3.8 | 55.1±3.4 | 52.6±4.3 | 60.3 |
| GMT | 67.4±1.0 | 61.7±2.1 | 55.8±0.7 | 57.0±2.4 | 60.2±0.5 | 58.2±2.0 | 57.8±3.2 | 65.7±1.3 | 60.2±0.3 | 57.3±2.3 | 60.7±0.6 | 57.1±1.2 | 59.9 |
| CIN | 70.4±2.8 | 66.5±4.3 | 58.5±2.6 | 64.2±2.7 | 60.6±3.0 | 64.2±3.2 | 58.7±2.4 | 69.1±2.7 | 57.5±3.4 | 67.7±2.1 | 59.5±2.3 | 56.1±1.2 | 62.7 |
| CDAN | 72.9±0.2 | 74.0±0.3 | 62.7±0.3 | 73.8±0.5 | 61.2±1.0 | 70.0±1.2 | 62.8±0.1 | 73.0±0.3 | 60.6±0.2 | 71.6±1.5 | 60.5±0.2 | 61.1±1.4 | 67.0 |
| ToAlign | 68.0±3.8 | 73.4±2.7 | 64.5±1.1 | 63.7±2.4 | 60.6±1.2 | 61.9±1.3 | 64.8±1.4 | 74.0±1.3 | 60.0±0.6 | 65.7±3.1 | 61.0±1.4 | 56.2±2.3 | 64.5 |
| MetaAlign | 73.6±0.2 | 72.7±1.9 | 63.9±1.0 | 67.9±4.3 | 60.4±0.7 | 65.4±1.8 | 65.2±0.8 | 73.2±2.3 | 60.0±0.6 | 66.7±2.4 | 61.2±1.1 | 56.8±2.1 | 65.6 |
| DEAL | 75.4±0.3 | 74.6±1.1 | 66.1±0.6 | 74.6±0.8 | 53.8±1.0 | 69.6±1.8 | 66.4±0.3 | 73.9±0.6 | 61.6±1.4 | 69.8±0.2 | 60.7±1.0 | 58.3±0.9 | 67.1 |
| CoCo | 74.6±0.9 | 77.2±0.6 | 64.1±3.4 | 73.8±1.1 | 60.5±0.2 | 71.5±0.7 | 65.9±0.5 | 76.0±0.5 | 61.4±0.4 | 72.6±0.6 | 59.6±1.0 | 64.7±1.0 | 68.5 |
| SGDA | 56.6±0.6 | 56.9±0.8 | 55.3±1.2 | 54.6±0.5 | 57.9±1.3 | 58.3±0.4 | 56.1±0.9 | 55.9±0.6 | 54.6±1.3 | 56.7±0.5 | 53.3±0.7 | 56.8±1.1 | 56.1 |
| DGDA | OOM | OOM | OOM | OOM | OOM | OOM | OOM | OOM | OOM | OOM | OOM | OOM | OOM |
| A2GNN | 55.4±0.8 | 56.1±0.6 | 56.7±1.0 | 55.3±0.5 | 54.9±0.7 | 57.2±0.9 | 55.7±0.5 | 56.5±1.3 | 54.5±0.6 | 56.8±0.5 | 56.2±1.0 | 58.8±0.8 | 56.1 |
| PA-BOTH | 56.1±0.5 | 56.0±0.4 | 56.3±0.7 | 56.4±0.4 | 56.0±0.6 | 57.1±0.7 | 56.2±1.1 | 58.3±0.9 | 56.5±0.6 | 57.2±0.9 | 56.9±0.4 | 57.7±0.8 | 56.8 |
| CoCA | **82.3±1.1** | **85.1±2.3** | **76.5±1.5** | **79.7±1.8** | **71.2±1.9** | **78.6±2.1** | **74.3±0.5** | **82.3±0.4** | **70.3±2.3** | **75.4±1.7** | **71.8±0.3** | **72.4±1.6** | **76.7** |

Table 6: The classification results (in %) on NCI1 under edge density domain shift (source→target). N0, N1, N2, and N3 denote the sub-datasets partitioned with edge density. **Bold** results indicate the best performance.

| Methods | N0→N1 | N1→N0 | N0→N2 | N2→N0 | N0→N3 | N3→N0 | N1→N2 | N2→N1 | N1→N3 | N3→N1 | N2→N3 | N3→N2 | Avg. |
|---|---|---|---|---|---|---|---|---|---|---|---|---|---|
| WL subtree | 72.6 | 80.3 | 62.7 | 75.5 | 52.0 | 63.6 | 69.1 | 69.8 | 70.7 | 59.4 | 80.0 | 70.6 | 68.9 |
| GCN | 49.5±0.4 | 71.1±0.4 | 46.8±0.5 | 33.7±2.8 | 32.7±0.4 | 27.4±0.1 | 56.2±1.5 | 55.3±0.4 | 58.2±1.7 | 51.0±0.2 | 60.7±3.7 | 53.2±0.2 | 49.6 |
| GIN | 67.3±2.7 | 67.9±4.8 | 61.5±4.2 | 65.4±3.7 | 58.9±4.1 | 61.0±3.4 | 62.5±3.2 | 66.2±2.1 | 69.7±0.9 | 56.8±0.7 | 72.4±2.8 | 64.0±1.6 | 64.5 |
| GMT | 50.3±1.2 | 42.5±3.4 | 51.1±3.7 | 42.5±4.5 | 56.1±4.7 | 42.5±4.1 | 53.2±4.9 | 51.0±0.2 | 68.2±0.4 | 51.0±0.3 | 68.2±0.5 | 53.2±0.4 | 52.5 |
| CIN | 51.1±0.2 | 72.6±0.1 | 54.0±0.9 | 72.6±0.2 | 68.2±0.3 | 71.5±1.3 | 55.0±2.1 | 53.5±1.8 | 68.2±0.3 | 52.0±0.3 | 68.3±0.1 | 53.6±0.6 | 61.7 |
| CDAN | 59.6±0.3 | 73.8±0.5 | 56.7±1.4 | 73.7±0.3 | 71.2±0.4 | 73.2±0.3 | 53.2±0.6 | 57.3±1.1 | 69.9±0.2 | 54.6±2.0 | 69.8±1.4 | 56.6±0.3 | 64.3 |
| ToAlign | 51.0±0.2 | 27.4±0.1 | 53.2±0.4 | 27.4±0.2 | 68.2±0.3 | 27.4±0.3 | 53.2±0.1 | 51.0±0.2 | 68.2±0.2 | 51.0±0.4 | 68.2±0.3 | 53.2±0.2 | 50.0 |
| MetaAlign | 65.0±0.7 | 77.6±1.6 | 62.0±0.6 | 77.1±0.9 | 68.2±0.8 | 74.5±2.0 | 64.2±0.9 | 65.4±0.3 | 68.0±0.3 | 56.1±2.3 | 68.2±0.1 | 66.2±1.1 | 67.7 |
| DEAL | 65.6±0.6 | 73.0±0.9 | 58.0±0.3 | 71.6±1.6 | 60.1±2.8 | 73.1±0.5 | 62.8±1.0 | 65.0±2.4 | 65.8±0.8 | 53.9±2.6 | 57.6±2.8 | 56.7±3.1 | 63.6 |
| CoCo | 70.4±0.7 | 80.4±0.9 | 62.4±0.8 | 75.8±1.2 | 65.7±2.0 | 73.7±0.3 | 67.0±0.8 | 70.4±0.7 | 69.7±0.4 | 62.7±0.9 | 74.4±0.5 | 63.7±0.9 | 69.7 |
| SGDA | OOM | OOM | OOM | OOM | OOM | OOM | OOM | OOM | OOM | OOM | OOM | OOM | OOM |
| DGDA | OOM | OOM | OOM | OOM | OOM | OOM | OOM | OOM | OOM | OOM | OOM | OOM | OOM |
| A2GNN | 59.2±0.8 | 58.7±0.5 | 59.0±1.1 | 58.7±0.8 | 58.9±0.6 | 59.2±1.2 | 58.7±0.6 | 58.6±1.2 | 59.0±1.0 | 59.5±0.6 | 58.7±0.5 | 58.5±1.1 | 58.9 |
| PA-BOTH | 57.6±0.5 | 58.4±0.4 | 58.9±0.6 | 57.4±0.6 | 57.1±1.0 | 58.4±0.5 | 58.0±1.0 | 58.1±0.5 | 58.4±0.6 | 57.7±1.1 | 57.5±0.6 | 58.0±0.4 | 58.0 |
| CoCA | **78.2±0.9** | **85.1±0.6** | **70.9±2.7** | **82.6±0.4** | **74.5±0.3** | **81.4±0.5** | **73.1±1.5** | **80.8±1.2** | **71.8±1.3** | **72.5±1.9** | **81.6±1.4** | **76.5±0.6** | **77.4** |

- **GMT** (Baek et al., 2021): GMT is a model based on multi-head attention, aiming to capture interactions between nodes based on their structural dependencies.

- **CIN** (Bodnar et al., 2021): CIN extends the theoretical findings from Simplicial Complexes to regular Cell Complexes, resulting in improved performance.

- **CDAN** (Long et al., 2018): CDAN employs an adversarial adaptation framework conditioned on discriminative information extracted from the classifier's predictions.

- **ToAlign** (Wei et al., 2021b): ToAlign strives to align the domain by conducting feature decomposition, incorporating prior knowledge such as the classification task itself.

- **MetaAlign** (Wei et al., 2021a): MetaAlign disentangles domain alignment and classification objectives into two distinct tasks, namely, meta-train and meta-test. It employs a meta-optimization approach to optimize both tasks.

- **DEAL** (Yin et al., 2022): DEAL employs adaptive perturbations that undergo adversarial training against a domain discriminator, aiming to address domain discrepancy.

- **CoCo** (Yin et al., 2023): CoCo leverages coupled branches and integrated contrastive learning techniques to mitigate domain discrepancies.

# G  MORE EXPERIMENTS

Table 6 and 7 shows the comparison performance of CoCA and baselines. From the results, we have a similar observation as we proposed in Section 5.2. Additionally, we utilize other large datasets (e.g. reddit_threads with 203,088 graphs) that align with the requirements of graph domain adaptation to validate our method. The results are reported in Table 14. From the results, we find that, the proposed CoCA still outperforms other baselines.

Table 7: The classification results (in %) on PROTEINS under edge density domain shift (source→target). P0, P1, P2, and P3 denote the sub-datasets partitioned with edge density. **Bold** results indicate the best performance.

| Methods | P0→P1 | P1→P0 | P0→P2 | P2→P0 | P0→P3 | P3→P0 | P1→P2 | P2→P1 | P1→P3 | P3→P1 | P2→P3 | P3→P2 | Avg. |
|---|---|---|---|---|---|---|---|---|---|---|---|---|---|
| WL subtree | 68.7 | 82.3 | 50.7 | 82.3 | 58.1 | 83.8 | 64.0 | 74.1 | 43.7 | 70.5 | 71.3 | 60.1 | 67.5 |
| GCN | 73.4±0.2 | 83.5±0.3 | 57.6±0.2 | 84.2±1.8 | 24.0±0.1 | 16.6±0.4 | 57.6±0.2 | 73.7±0.4 | 24.0±0.1 | 26.6±0.2 | 39.9±0.9 | 42.5±0.1 | 50.3 |
| GIN | 62.5±4.7 | 74.9±3.7 | 53.0±4.6 | 59.6±4.2 | 73.7±0.8 | 64.7±3.4 | 60.6±2.7 | 69.8±0.6 | 31.1±2.8 | 63.1±3.4 | 72.3±2.7 | 64.6±1.4 | 62.5 |
| GMT | 73.4±0.3 | 83.5±0.2 | 57.6±0.1 | 83.5±0.3 | 24.0±0.1 | 83.5±0.1 | 57.4±0.2 | 73.4±0.2 | 24.1±0.1 | 73.4±0.3 | 24.0±0.1 | 57.6±0.2 | 59.6 |
| CIN | 74.5±0.2 | 84.1±0.5 | 57.8±0.2 | 82.7±0.9 | 75.6±0.6 | 79.2±2.2 | 61.5±2.7 | 74.0±1.0 | 75.5±0.8 | 72.5±2.1 | 76.0±0.3 | 60.9±1.2 | 72.9 |
| CDAN | 72.2±1.8 | 82.4±1.6 | 59.8±2.1 | 76.8±2.4 | 69.3±4.1 | 71.8±3.7 | 64.4±2.5 | 74.3±0.4 | 46.3±2.0 | 69.8±1.8 | 74.4±1.7 | 62.6±2.3 | 68.7 |
| ToAlign | 73.4±0.1 | 83.5±0.2 | 57.6±0.1 | 83.5±0.2 | 24.0±0.3 | 83.5±0.4 | 57.6±0.1 | 73.4±0.1 | 24.0±0.2 | 73.4±0.2 | 24.0±0.1 | 57.6±0.3 | 59.6 |
| MetaAlign | 75.5±0.9 | 84.9±0.6 | 64.8±1.6 | 85.9±1.1 | 69.3±2.7 | 83.3±0.6 | 68.7±1.2 | 74.2±0.7 | 73.3±3.3 | 72.2±0.9 | 69.9±1.8 | 63.6±2.3 | 73.8 |
| DEAL | 76.5±0.4 | 83.1±0.4 | **67.5±1.3** | 77.6±1.8 | 76.0±0.2 | 80.1±2.7 | 66.1±1.3 | 75.4±1.5 | 42.3±4.1 | 68.1±3.7 | 73.1±2.2 | 67.8±1.2 | 71.1 |
| CoCo | 75.5±0.2 | 84.2±0.4 | 59.8±0.5 | 83.4±0.2 | 73.6±2.3 | 81.6±2.4 | 65.8±0.3 | 76.2±0.2 | 75.8±0.2 | 71.1±2.1 | 76.1±0.2 | 67.1±0.6 | 74.2 |
| SGDA | 63.8±0.6 | 65.2±1.3 | 66.7±1.0 | 59.1±1.5 | 60.1±0.8 | 64.4±1.2 | 65.2±0.7 | 63.9±0.9 | 64.5±0.6 | 61.1±1.3 | 58.9±1.4 | 64.9±1.2 | 63.2 |
| DGDA | 58.7±0.8 | 59.9±1.2 | 57.1±0.6 | 57.9±0.8 | 59.2±1.3 | 58.9±0.4 | 61.1±1.2 | 60.3±1.6 | 58.6±0.9 | 57.5±1.2 | 58.4±0.5 | 62.3±1.5 | 59.2 |
| A2GNN | 65.4±1.3 | 66.3±1.1 | 68.2±1.4 | 66.3±1.2 | 65.4±0.7 | 65.9±0.9 | 66.9±1.3 | 65.4±1.2 | 65.6±0.9 | 65.5±1.2 | 66.1±2.0 | 66.0±1.8 | 66.1 |
| PA-BOTH | 63.1±0.7 | 67.2±1.1 | 64.3±0.5 | 72.1±1.8 | 66.3±0.7 | 64.1±1.2 | 69.9±2.1 | 67.5±1.8 | 61.2±1.4 | 67.7±2.3 | 61.2±1.6 | 65.5±0.6 | 65.9 |
| CoCA | **76.9±0.5** | **88.4±2.2** | 67.2±1.3 | **88.4±1.6** | **77.0±0.9** | **88.4±1.3** | **69.4±0.8** | **76.5±1.5** | **78.8±1.2** | **76.2±1.8** | **78.4±0.6** | **70.8±1.1** | **78.0** |

Table 8: The graph classification results (in %) on Mutagenicity under node domain shift (source→target). P0, P1, P2, and P3 denote the sub-datasets partitioned with node. **Bold** results indicate the best performance.

| Methods | M0→M1 | M1→M0 | M0→M2 | M2→M0 | M0→M3 | M3→M0 | M1→M2 | M2→M1 | M1→M3 | M3→M1 | M2→M3 | M3→M2 | Avg. |
|---|---|---|---|---|---|---|---|---|---|---|---|---|---|
| WL subtree | **78.0** | 68.7 | 70.1 | 70.5 | 59.0 | 61.2 | 71.7 | 78.0 | 49.9 | 56.3 | 69.4 | 71.9 | 67.1 |
| GCN | 74.5±0.2 | 60.8±2.1 | 69.7±0.4 | 68.5±1.7 | 54.1±0.9 | 55.2±0.9 | 68.6±1.6 | 75.5±0.5 | 51.5±1.3 | 46.4±1.7 | 58.6±0.4 | 60.2±0.2 | 61.9 |
| GIN | 77.9±3.1 | 70.7±2.4 | 70.9±0.8 | 69.2±1.2 | 64.1±1.0 | 61.9±2.4 | 78.5±0.2 | **79.8±3.3** | 65.5±2.7 | 71.5±0.9 | 69.5±1.8 | 73.5±2.6 | 71.1 |
| GMT | 67.3±0.2 | 52.5±0.1 | 59.9±0.3 | 47.5±0.2 | 53.5±0.2 | 52.5±0.4 | 67.3±0.2 | 46.7±0.5 | 67.3±0.3 | 53.3±0.1 | 59.9±0.4 |  | 57.1 |
| CIN | 70.8±1.1 | 66.9±3.4 | 61.7±0.6 | 62.6±2.4 | 56.3±3.1 | 62.9±1.3 | 65.1±1.0 | 68.8±1.7 | 56.6±1.4 | 66.9±1.0 | 58.1±1.3 | 62.5±0.9 | 63.3 |
| CDAN | 75.5±0.1 | 71.3±0.4 | 70.7±0.3 | 70.3±0.1 | 58.7±0.6 | 58.4±0.6 | 70.2±0.5 | 76.1±0.5 | 58.5±0.6 | 69.4±1.5 | 59.0±0.1 | 63.7±1.4 | 66.8 |
| ToAlign | 67.3±0.2 | 47.5±0.4 | 59.9±0.6 | 47.5±0.5 | 46.7±0.4 | 47.5±0.2 | 59.9±0.7 | 67.3±0.3 | 46.7±0.1 | 67.3±0.4 | 46.7±0.5 | 59.9±0.3 | 55.4 |
| MetaAlign | 76.5±0.4 | 71.8±1.1 | 71.8±0.6 | 71.4±0.9 | 59.3±0.8 | 63.0±1.0 | 74.2±1.4 | 78.0±0.2 | 61.7±1.2 | 69.9±1.6 | 62.2±0.4 | 68.3±1.5 | 69.0 |
| DEAL | 76.6±0.8 | 68.8±1.0 | 69.9±0.4 | 66.4±0.8 | 59.3±2.1 | 64.2±2.2 | 79.1±0.1 | 81.9±0.6 | 64.5±1.1 | 75.3±0.6 | 69.8±1.6 | 76.5±0.2 | 71.0 |
| CoCo | 75.5±0.4 | 71.7±0.7 | 68.7±1.1 | 69.2±2.0 | 60.8±1.1 | 65.7±0.3 | **79.2±1.2** | 76.8±0.6 | 63.8±0.5 | 73.8±0.4 | 64.6±0.8 | 70.1±1.1 | 70.0 |
| SGDA | OOM | OOM | OOM | OOM | OOM | OOM | OOM | OOM | OOM | OOM | OOM | OOM | OOM |
| DGDA | OOM | OOM | OOM | OOM | OOM | OOM | OOM | OOM | OOM | OOM | OOM | OOM | OOM |
| A2GNN | 55.4±0.6 | 56.3±0.2 | 55.6±0.8 | 55.1±0.5 | 55.3±1.1 | 55.9±0.4 | 56.1±0.7 | 55.7±0.6 | 57.1±0.3 | 56.6±1.2 | 55.2±0.7 | 56.8±1.0 | 55.9 |
| PA-BOTH | 55.9±1.0 | 56.0±0.5 | 56.1±0.7 | 56.6±1.2 | 55.9±0.6 | 56.0±0.7 | 57.3±0.8 | 56.8±1.3 | 55.9±1.2 | 56.3±1.0 | 56.4±0.9 | 57.1±1.3 | 56.4 |
| CoCA | 82.9±1.3 | 77.5±1.7 | 79.3±0.5 | 77.3±0.7 | 77.4±1.1 | 74.2±0.9 | 82.1±0.4 | 86.1±1.0 | 77.7±0.8 | 81.3±1.2 | 78.5±0.3 | 81.6±0.7 | 79.7 |

# H MORE FLEXIBILITY EXPERIMENTS

Figure 5 depicts the comparative results of NCI1 and PROTEINS datasets across different Graph Neural Network (GNN) models and various kernel methods.

# I CASE STUDY

To further elucidate the effective principles of the CoCA model and provide a more intuitive understanding of its operational mechanism, we showcase the accuracy of pseudo-label data filtered during the branch coupling module. The results are illustrated in Figure 6, where solid lines denote the accuracy of pseudo-labels filtered from the MP branch, and dashed lines represent the results during the SP branch. From Figure 6, we have the following observations: (1) As the iterations of the filtering process progress, the accuracy of the pseudo-labeled data for both branches shows an upward trend. This is intuitively understandable, as the quality of the filtered samples improves, enhancing the predictive capabilities of the model in subsequent steps, thus creating a positive feedback loop. (2) In most scenarios, the accuracy of the filtered samples during both the MP and SP branches exhibits a similar changing trend, indicating mutual influence and reinforcement between the two branches. (3) Comparatively, the accuracy of samples filtered during the MP branch is noticeably higher than that of the SP branch. This difference could be attributed to the powerful information retrieval capability of the GMT employed in the MP branch.

Additionally, we include experiments with GMT as the backbone and demonstrate that our method still outperforms others under the same backbone, we have ensured a fair and robust comparison. The results are shown in Table 15. From the results, we can validate the superiority of the proposed CoCA.

Table 9: The graph classification results (in %) on NCI1 under node domain shift (source→target). P0, P1, P2, and P3 denote the sub-datasets partitioned with node. **Bold** results indicate the best performance.

| Methods | N0→N1 | N1→N0 | N0→N2 | N2→N0 | N0→N3 | N3→N0 | N1→N2 | N2→N1 | N1→N3 | N3→N1 | N2→N3 | N3→N2 | Avg. |
|---|---|---|---|---|---|---|---|---|---|---|---|---|---|
| WL subtree | 73.5 | 79.5 | 64.8 | 75.9 | 58.9 | 68.4 | 72.5 | 72.0 | 69.7 | 63.6 | 76.1 | 74.0 | 70.7 |
| GCN | 51.2±0.1 | 71.1±0.4 | 42.7±0.4 | 27.8±0.3 | 32.1±1.1 | 27.0±0.2 | 55.2±0.6 | 50.5±0.7 | 50.9±1.1 | 49.1±0.3 | 67.1±0.6 | 57.3±0.6 | 48.5 |
| GIN | 66.9±2.2 | 78.9±2.3 | 60.3±3.1 | 72.8±0.3 | 51.1±0.6 | 68.6±1.8 | 63.5±2.1 | 67.8±3.7 | 65.9±1.7 | 71.1±1.1 | 67.2±1.3 | | 66.2 |
| GMT | 50.9±0.5 | 73.0±0.1 | 57.3±0.3 | 73.0±0.4 | 66.5±0.2 | 73.0±0.3 | 72.4±0.6 | 50.9±0.1 | 66.5±0.4 | 58.3±0.2 | 66.5±0.5 | 72.8±0.3 | 65.1 |
| CIN | 60.1±0.7 | 73.1±1.1 | 57.5±0.2 | 73.0±0.4 | 66.5±1.1 | 73.1±0.7 | 58.5±2.1 | 52.9±1.4 | 66.5±1.3 | 56.1±0.1 | 66.5±0.4 | 57.4±0.7 | 63.4 |
| CDAN | 57.1±0.4 | 75.0±0.7 | 61.2±0.4 | 73.7±0.1 | 68.2±0.4 | 73.3±0.3 | 60.2±0.1 | 56.5±1.4 | 68.2±0.2 | 53.9±1.4 | 68.4±0.2 | 59.6±0.5 | 64.6 |
| ToAlign | 49.1±0.3 | 77.0±0.2 | 57.3±0.5 | 27.0±0.4 | 66.5±0.5 | 27.0±0.2 | 57.3±0.3 | 49.1±0.4 | 66.5±0.2 | 49.1±0.3 | 66.5±0.1 | 57.3±0.4 | 50.0 |
| MetaAlign | 65.6±1.8 | 77.7±0.2 | 63.5±1.4 | 75.7±0.7 | 66.4±0.3 | 74.0±0.3 | 66.3±1.1 | 64.6±1.2 | 66.7±0.2 | 59.5±2.6 | 66.7±0.3 | 66.7±2.7 | 67.8 |
| DEAL | 64.0±0.9 | 71.9±1.2 | 61.4±0.3 | 73.3±0.4 | 64.9±1.4 | 71.9±1.9 | 62.5±2.1 | 66.2±0.5 | 54.2±1.4 | 55.6±0.8 | 64.6±0.4 | 58.8±0.4 | 64.1 |
| CoCo | 69.7±0.1 | **80.4±0.4** | 64.7±1.2 | **76.5±0.4** | 65.0±1.7 | 73.9±0.3 | 68.9±1.3 | 70.7±0.9 | 68.2±1.2 | 61.4±1.7 | 73.0±0.1 | 65.2±0.9 | 69.8 |
| SGDA | OOM | OOM | OOM | OOM | OOM | OOM | OOM | OOM | OOM | OOM | OOM | OOM | OOM |
| DGDA | OOM | OOM | OOM | OOM | OOM | OOM | OOM | OOM | OOM | OOM | OOM | OOM | OOM |
| A2GNN | 59.0±0.6 | 58.3±1.1 | 58.5±0.8 | 58.6±1.3 | 58.7±1.0 | 59.0±0.7 | 58.5±1.1 | 58.7±1.5 | 59.1±0.6 | 58.3±1.2 | 58.6±0.7 | 59.0±0.5 | 58.7 |
| PA-BOTH | 57.7±0.4 | 58.0±0.6 | 57.9±0.5 | 56.9±0.8 | 57.4±0.6 | 58.3±0.5 | 57.1±1.2 | 58.8±0.9 | 58.1±0.7 | 58.0±0.9 | 57.9±0.5 | 58.3±0.8 | 57.9 |
| CoCA | **82.4±0.7** | **85.9±1.0** | **70.4±0.6** | **80.4±1.5** | **74.2±0.5** | **86.3±0.7** | **79.1±1.1** | **78.4±1.6** | **75.5±0.5** | **81.9±1.7** | **80.9±1.4** | **75.2±0.8** | **79.2** |

Table 10: The graph classification results (in %) on PROTEINS under node domain shift (source→target). P0, P1, P2, and P3 denote the sub-datasets partitioned with node. **Bold** results indicate the best performance.

| Methods | P0→P1 | P1→P0 | P0→P2 | P2→P0 | P0→P3 | P3→P0 | P1→P2 | P2→P1 | P1→P3 | P3→P1 | P2→P3 | P3→P2 | Avg. |
|---|---|---|---|---|---|---|---|---|---|---|---|---|---|
| WL subtree | 69.1 | 59.7 | 61.2 | 75.9 | 41.6 | 83.5 | 57.6 | 72.7 | 24.7 | 72.7 | 63.1 | 62.9 | 62.4 |
| GCN | 73.7±0.3 | 82.7±0.4 | 57.6±0.2 | 84.0±1.3 | 24.4±0.4 | 17.3±0.2 | 57.6±0.1 | 70.9±0.7 | 24.4±0.5 | 26.3±0.1 | 37.5±0.2 | 42.5±0.8 | 49.9 |
| GIN | 71.8±2.7 | 70.2±4.7 | 58.5±4.3 | 56.9±4.9 | 74.2±1.7 | 78.2±3.3 | 63.3±2.7 | 67.1±3.8 | 35.9±4.2 | 61.0±2.4 | 71.9±2.1 | 65.1±1.0 | 64.5 |
| GMT | 73.7±0.2 | 82.7±0.1 | 57.6±0.3 | 78.6±3.1 | 75.6±1.4 | 17.3±0.6 | 57.6±1.5 | 74.1±0.6 | 73.7±0.6 | 75.6±0.4 | 26.3±1.2 | 42.4±0.5 | 61.8 |
| CIN | 74.1±0.6 | 83.8±1.0 | 60.1±2.1 | 78.6±3.1 | 75.6±0.2 | 74.8±3.7 | 63.9±2.7 | 74.1±0.6 | 57.0±4.3 | 58.9±3.3 | 75.6±0.7 | 63.6±1.0 | 70.0 |
| CDAN | 75.9±1.0 | 83.1±0.6 | 60.8±0.6 | 82.6±0.2 | 75.8±0.3 | 70.9±2.4 | 64.7±0.3 | **77.7±0.6** | 73.3±1.8 | 75.4±0.7 | 75.8±0.4 | 67.1±0.8 | 73.6 |
| ToAlign | 73.7±0.4 | 82.7±0.3 | 57.6±0.6 | 82.7±0.8 | 24.4±0.1 | 82.7±0.3 | 57.6±0.4 | 73.7±0.2 | 24.4±0.7 | 73.7±0.3 | 24.4±0.5 | 57.6±0.4 | 59.6 |
| MetaAlign | 74.3±0.8 | 83.3±2.2 | 60.6±1.7 | 71.2±2.1 | 76.3±0.3 | 77.3±2.4 | 64.6±1.2 | 72.0±1.0 | 76.0±0.5 | 73.3±1.8 | 74.4±1.7 | 56.9±1.4 | 71.7 |
| DEAL | 75.4±1.2 | 78.0±2.4 | 68.1±1.9 | 80.8±2.1 | 73.8±1.4 | 80.6±2.3 | 65.7±1.7 | 74.7±2.4 | 74.7±1.6 | 71.0±2.1 | 68.1±2.6 | 70.3±0.4 | 73.4 |
| CoCo | 74.8±0.6 | 84.1±1.1 | 65.5±0.4 | 83.6±1.1 | 72.4±2.9 | 83.1±0.4 | 69.7±0.5 | 75.8±0.7 | 71.4±2.3 | 73.4±1.3 | 72.5±2.7 | 66.4±1.7 | 74.4 |
| SGDA | 64.2±0.5 | 61.0±0.7 | 66.9±1.2 | 61.9±0.9 | 65.4±1.6 | 66.5±1.0 | 64.6±1.1 | 60.1±0.5 | 66.3±1.3 | 59.3±0.8 | 66.0±1.6 | 66.2±1.3 | 64.1 |
| DGDA | 58.1±0.4 | 58.6±0.6 | 58.9±1.0 | 61.0±0.9 | 59.6±0.7 | 60.2±1.5 | 56.7±0.6 | 56.8±0.8 | 58.1±0.4 | 58.8±1.1 | 57.0±1.2 | 62.2±1.6 | 58.9 |
| A2GNN | 65.7±0.6 | 65.9±0.8 | 66.3±0.9 | 65.6±1.1 | 65.2±1.4 | 65.6±1.3 | 65.9±1.7 | 65.8±1.6 | 65.0±1.5 | 66.1±1.2 | 65.2±1.9 | 65.9±1.8 | 65.7 |
| PA-BOTH | 61.0±0.8 | 61.2±1.3 | 60.3±0.6 | 66.7±2.1 | 63.7±1.5 | 61.9±2.0 | 66.2±1.4 | 69.9±2.3 | 68.0±0.7 | 69.4±1.8 | 61.5±0.4 | 67.6±1.0 | 64.9 |
| CoCA | **77.6±0.9** | **85.3±1.1** | **70.5±0.6** | **84.8±1.4** | **76.6±0.7** | **83.9±0.9** | **71.9±0.6** | 76.9±1.1 | **76.1±0.8** | 73.7±1.0 | **77.0±1.2** | **72.3±0.7** | **77.2** |

## J  IMPACT STATEMENTS

This work introduces an innovative approach for unsupervised graph domain adaptation, with the objective of advancing the machine learning field, particularly in the domain of transfer learning. The proposed method has the potential to substantially enhance the efficiency and scalability of transfer learning tasks. The societal implications of this research are multifaceted. The introduced method has the capacity to contribute to the development of more efficient and effective machine learning systems, with potential applications across various domains, including healthcare, education, and technology. Such advancements could lead to improved services and products, ultimately benefiting society as a whole.

Table 11: The graph classification results (in %) on Mutagenicity under graph flux domain shift (source→target). M0, M1, M2, and M3 denote the sub-datasets partitioned with graph flux. **Bold** results indicate the best performance. OOM means out of memory.

| Methods | M0→M1 | M1→M0 | M0→M2 | M2→M0 | M0→M3 | M3→M0 | M1→M2 | M2→M1 | M1→M3 | M3→M1 | M2→M3 | M3→M2 | Avg. |
|---|---|---|---|---|---|---|---|---|---|---|---|---|---|
| WL subtree | 74.4 | 72.9 | 64.9 | 68.9 | 49.1 | 59.8 | 70.0 | 70.5 | 76.9 | 60.7 | 82.6 | 70.5 | 68.5 |
| GCN | 63.1±1.0 | 68.1±0.3 | 48.8±0.4 | 62.6±0.3 | 29.1±2.1 | 38.8±0.3 | 54.3±0.1 | 61.8±0.5 | 30.4±0.2 | 43.6±0.3 | 67.8±0.1 | 57.9±1.3 | 52.2 |
| GIN | 68.1±1.6 | 74.2±0.6 | 59.6±2.3 | 65.2±1.4 | 40.3±2.7 | 54.6±1.8 | 61.3±1.1 | 63.1±3.2 | 71.6±3.0 | 60.0±1.4 | 79.7±1.3 | 69.2±0.7 | 63.9 |
| GMT | 56.5±0.3 | 60.7±0.4 | 57.9±0.2 | 40.2±1.2 | 80.6±0.4 | 39.3±0.6 | 57.9±1.1 | 45.0±2.1 | 80.6±0.5 | 43.5±1.1 | 80.6±1.4 | 57.9±2.2 | 58.4 |
| CIN | 64.1±3.0 | 61.3±0.5 | 63.5±2.3 | 63.6±1.5 | 78.2±0.5 | 63.9±2.7 | 60.6±1.5 | 57.0±0.4 | 73.7±3.2 | 61.4±1.0 | 79.1±2.1 | 61.1±1.9 | 65.6 |
| CDAN | 62.8±0.3 | 68.2±0.6 | 63.6±0.6 | 66.9±1.7 | 81.2±0.5 | 65.0±2.1 | 65.8±0.2 | 64.7±1.2 | 80.7±0.1 | 62.5±2.3 | 82.4±0.4 | 66.0±0.5 | 69.1 |
| ToAlign | 43.5±0.4 | 39.3±0.7 | 57.9±1.0 | 39.3±1.4 | 80.6±1.1 | 39.3±0.7 | 57.9±0.3 | 43.5±2.1 | 80.6±1.8 | 43.5±0.4 | 80.6±0.9 | 57.9±1.0 | 55.3 |
| MetaAlign | 63.1±2.5 | 68.8±2.6 | 63.3±0.6 | 65.2±2.2 | 81.9±0.1 | 64.5±1.4 | 65.0±0.6 | 68.3±0.6 | 81.0±0.3 | 65.2±0.2 | 82.5±0.4 | 68.3±0.6 | 69.7 |
| 0 DEAL | 64.6±0.5 | 65.5±0.8 | 64.2±1.0 | 63.1±2.1 | 82.7±0.8 | 62.8±0.7 | 70.2±0.4 | 67.3±0.4 | 79.6±0.1 | 63.9±1.4 | 75.7±0.3 | 67.0±0.2 | 68.9 |
| CoCo | 65.7±1.8 | 74.1±0.7 | 65.1±0.2 | 67.6±0.9 | 80.5±1.3 | 56.5±1.7 | 68.4±1.3 | 70.7±0.4 | 78.9±1.2 | 67.3±0.3 | 83.7±0.1 | 71.5±0.9 | 070.8 |
| SGDA | OOM | OOM | OOM | OOM | OOM | OOM | OOM | OOM | OOM | OOM | OOM | OOM | OOM |
| DGDA | OOM | OOM | OOM | OOM | OOM | OOM | OOM | OOM | OOM | OOM | OOM | OOM | OOM |
| A2GNN | 55.4±0.3 | 55.7±0.7 | 55.6±0.5 | 54.7±0.8 | 63.3±1.0 | 56.6±0.9 | 55.3±0.6 | 55.7±0.5 | 65.5±0.8 | 56.6±1.2 | 69.9±1.4 | 55.0±0.5 | 58.3 |
| PA-BOTH | 55.9±0.9 | 56.0±0.4 | 56.4±0.8 | 56.2±0.5 | 67.1±1.2 | 59.8±0.7 | 57.3±0.9 | 56.0±1.2 | 69.9±1.5 | 58.0±0.8 | 67.5±0.9 | 56.4±1.0 | 59.7 |
| CoCA | **77.1±0.7** | **79.7±1.0** | **75.5±0.5** | **77.3±1.7** | **84.9±1.5** | **75.1±1.1** | **77.6±1.2** | **75.7±0.8** | **84.1±2.0** | **74.4±0.9** | **86.2±0.5** | **78.4±0.8** | **78.8** |

Table 12: The graph classification results (in %) on FRANKENSTEIN under graph flux domain shift (source→target). F0, F1, F2, and F3 denote the sub-datasets partitioned with graph flux. **Bold** results indicate the best performance. OOM means out of memory.

| Methods | F0→F1 | F1→F0 | F0→F2 | F2→F0 | F0→F3 | F3→F0 | F1→F2 | F2→F1 | F1→F3 | F3→F1 | F2→F3 | F3→F2 | Avg. |
|---|---|---|---|---|---|---|---|---|---|---|---|---|---|
| WL subtree | 58.4 | 51.8 | 58.7 | 51.3 | 64.3 | 48.9 | 64.9 | 58.9 | 78.5 | 54.6 | 57.1 | 61.3 | 59.1 |
| GCN | 56.2±0.2 | 59.0±1.3 | 41.4±0.4 | 45.8±0.5 | 21.2±0.7 | 41.4±1.7 | 42.5±1.6 | 49.0±0.2 | 24.1±1.6 | 44.8±0.7 | 81.4±0.3 | 58.8±0.2 | 47.1 |
| GIN | 60.7±0.6 | 58.0±1.0 | 61.0±2.3 | 58.9±2.3 | 77.5±2.2 | 45.3±2.5 | 62.5±2.0 | 59.2±3.0 | 71.4±2.8 | 49.8±1.7 | 77.9±1.4 | 59.9±0.5 | 61.8 |
| GMT | 56.2±0.4 | 59.8±0.2 | 41.4±0.3 | 59.8±0.7 | 21.2±1.1 | 59.8±0.5 | 41.4±0.2 | 56.2±0.2 | 21.1±1.1 | 56.2±1.4 | 78.8±0.6 | 58.6±0.8 | 50.9 |
| CIN | 57.8±1.1 | 60.1±0.7 | 58.6±0.2 | 59.8±0.2 | 78.9±0.1 | 59.9±0.4 | 58.8±0.3 | 57.4±0.5 | 78.8±0.6 | 57.7±1.2 | 78.8±0.7 | 60.1±1.1 | 63.9 |
| CDAN | 60.9±0.7 | 59.8±0.5 | 61.1±1.3 | 61.0±0.2 | 80.5±1.2 | 59.8±0.3 | 64.0±0.4 | 61.4±0.1 | 81.8±0.1 | 58.0±1.2 | 81.8±0.3 | 63.8±0.7 | 66.1 |
| ToAlign | 56.2±0.2 | 59.8±0.2 | 41.4±0.1 | 59.8±0.2 | 21.1±0.3 | 59.8±0.7 | 41.4±1.1 | 56.2±1.2 | 21.1±0.4 | 56.2±0.6 | 21.1±1.3 | 41.4±0.5 | 44.6 |
| MetaAlign | 57.3±2.4 | 59.1±1.1 | 60.9±1.5 | 60.2±0.4 | 80.3±2.1 | 60.4±0.6 | 64.0±1.1 | 64.9±0.6 | 81.4±1.2 | 58.5±2.3 | 80.8±0.5 | 63.4±1.8 | 65.9 |
| DEAL | 65.3±0.6 | 64.0±0.2 | 61.3±0.6 | 61.0±0.9 | 78.3±2.1 | 55.5±1.8 | 64.9±1.2 | 64.8±1.1 | 80.1±1.3 | 60.1±2.1 | 81.8±0.4 | 65.7±0.7 | 66.9 |
| CoCo | 63.5±2.4 | 61.5±1.0 | 64.4±1.0 | 61.2±0.7 | 81.7±0.4 | 55.0±1.6 | 64.5±0.6 | 64.6±1.1 | 80.4±1.5 | 60.6±1.5 | 81.5±0.6 | 62.2±1.7 | 66.8 |
| SGDA | 55.7±0.5 | 55.4±0.9 | 54.8±0.3 | 55.3±0.7 | 56.1±0.5 | 55.4±0.8 | 53.2±1.1 | 55.1±0.6 | 58.4±0.4 | 55.3±0.5 | 57.7±1.0 | 54.9±0.6 | 55.7 |
| DGDA | OOM | OOM | OOM | OOM | OOM | OOM | OOM | OOM | OOM | OOM | OOM | OOM | OOM |
| A2GNN | 56.0±0.3 | 56.3±0.6 | 55.6±0.4 | 57.3±0.7 | 58.6±0.6 | 55.9±0.9 | 55.5±0.5 | 55.3±0.2 | 62.1±1.3 | 56.6±0.9 | 65.5±0.8 | 56.0±1.0 | 57.5 |
| PA-BOTH | 56.3±0.5 | 56.9±0.7 | 56.4±0.6 | 59.9±1.0 | 60.3±1.3 | 56.2±1.0 | 57.7±0.4 | 56.6±0.8 | 66.7±0.9 | 58.3±1.2 | 69.9±1.5 | 59.0±0.6 | 59.6 |
| CoCA | **74.2±1.3** | **75.6±1.4** | **78.2±0.6** | **75.7±0.5** | **86.1±1.2** | **79.4±1.2** | **78.1±0.3** | **82.4±1.3** | **85.7±1.0** | **78.7±1.9** | **86.5±0.9** | **81.8±2.8** | **80.2** |

Table 13: The graph classification results (in %) on PROTEINS under graph flux domain shift (source→target). P0, P1, P2, and P3 denote the sub-datasets partitioned with graph flux. **Bold** results indicate the best performance.

| Methods | P0→P1 | P1→P0 | P0→P2 | P2→P0 | P0→P3 | P3→P0 | P1→P2 | P2→P1 | P1→P3 | P3→P1 | P2→P3 | P3→P2 | Avg. |
|---|---|---|---|---|---|---|---|---|---|---|---|---|---|
| WL subtree | 73.4 | 72.7 | 70.5 | 73.0 | 72.8 | 59.0 | 66.5 | 71.6 | 60.6 | 58.3 | 76.3 | 64.0 | 68.2 |
| GCN | 57.2±2.7 | 62.8±1.7 | 67.6±0.5 | 58.5±1.3 | 67.7±0.4 | 61.0±0.3 | 65.0±0.8 | 51.1±1.3 | 65.6±2.2 | 55.4±0.4 | 68.5±3.1 | 67.7±0.5 | 62.3 |
| GIN | 69.3±2.3 | 65.8±0.8 | 69.3±1.7 | 69.8±1.6 | 71.4±2.1 | 52.4±1.8 | 64.0±2.4 | 65.7±3.2 | 53.4±3.7 | 58.1±0.8 | 72.6±0.3 | 64.6±2.3 | 64.7 |
| GMT | 67.8±1.3 | 69.6±0.7 | 74.5±0.5 | 67.6±2.5 | 69.9±2.1 | 55.8±0.7 | 74.8±1.4 | 60.1±2.4 | 71.4±3.3 | 51.5±0.5 | 69.0±0.5 | 63.3±1.3 | 66.3 |
| CIN | 62.6±0.5 | 59.4±0.5 | 64.0±0.9 | 58.5±1.8 | 71.9±1.7 | 60.6±2.1 | 63.7±0.5 | 61.2±2.1 | 73.2±0.5 | 57.7±3.0 | 68.1±0.4 | 58.5±2.7 | 63.3 |
| CDAN | 75.6±0.5 | 70.5±0.6 | 71.6±0.5 | 69.8±0.5 | 76.6±0.8 | 71.4±0.3 | 71.4±0.3 | 72.1±0.3 | 75.5±0.7 | 74.3±0.8 | 78.2±1.1 | 74.0±0.8 | 73.4 |
| ToAlign | 51.1±0.6 | 55.8±0.1 | 63.3±0.2 | 63.5±0.4 | 68.1±0.7 | 55.8±0.3 | 63.3±0.5 | 51.1±0.2 | 68.1±1.0 | 51.1±0.4 | 68.1±0.6 | 63.3±0.2 | 59.6 |
| MetaAlign | 59.4±1.1 | 62.2±1.0 | 68.9±0.3 | 65.3±0.8 | 75.1±0.7 | 67.5±2.1 | 70.9±1.4 | 60.6±2.3 | 72.4±1.4 | 59.4±0.6 | 74.6±0.7 | 67.8±1.3 | 67.0 |
| DEAL | **76.6±0.4** | 62.8±0.8 | 72.8±1.3 | 67.3±2.2 | 77.2±2.3 | 67.6±1.9 | 71.2±1.6 | 56.0±2.5 | 73.9±2.1 | 66.0±0.3 | 76.4±1.1 | 65.5±2.1 | 69.4 |
| CoCo | 73.4±0.5 | 73.6±0.8 | 73.4±1.0 | 71.6±0.5 | 75.2±1.6 | 74.6±0.3 | 70.7±0.8 | 68.4±1.5 | 75.0±0.2 | 72.7±0.4 | 76.3±1.1 | **75.0±1.8** | 73.3 |
| SGDA | 63.8±0.8 | 65.2±0.5 | 66.7±0.3 | 59.1±1.2 | 62.3±0.7 | 60.6±0.4 | 65.2±0.9 | 61.8±1.0 | 64.5±1.3 | 60.9±0.8 | 59.4±1.2 | 64.9±1.1 | 62.9 |
| DGDA | 59.4±0.7 | 62.3±1.1 | 63.1±0.5 | 61.2±0.9 | 60.4±0.6 | 58.8±1.0 | 60.3±0.8 | 63.5±1.2 | 61.9±0.8 | 60.4±1.6 | 64.2±1.3 | 62.6±1.4 | 61.5 |
| A2GNN | 65.4±0.7 | 66.4±1.1 | 65.7±1.3 | 66.0±0.6 | 64.9±1.2 | 65.8±1.6 | 65.5±1.8 | 66.0±1.4 | 65.8±2.1 | 65.6±1.9 | 66.1±1.7 | 66.0±2.0 | 65.8 |
| PA-BOTH | 66.9±0.5 | 67.1±0.8 | 67.3±1.1 | 65.8±0.7 | 69.1±1.0 | 66.1±1.4 | 66.7±1.3 | 67.4±1.4 | 66.3±1.8 | 66.0±1.2 | 66.8±0.8 | 66.3±1.5 | 66.8 |
| CoCA | 75.8±0.2 | **76.6±0.5** | 74.8±1.3 | **79.1±0.9** | **80.2±0.6** | **78.0±1.2** | **75.1±0.5** | **76.2±1.0** | **79.9±0.9** | **78.3±0.6** | **80.2±1.6** | 74.4±0.6 | **77.4** |

Table 14: The classification results (in %) on reddit_threads under edge density domain shift (source→target). R0, R1, R2 denote the sub-datasets partitioned with edge density. Bold results indicate the best performance.

| Methods | R0→R1 | R1→R0 | R0→R2 | R2→R0 | R1→R2 | R2→R1 | Avg. |
|---|---|---|---|---|---|---|---|
| CDAN | 68.4±1.4 | 69.7±1.2 | 66.3±1.0 | 64.5±1.1 | 69.2±1.2 | 70.7±1.1 | 68.1 |
| DEAL | 70.7±1.3 | 71.4±1.0 | 70.5±1.0 | 71.3±0.9 | 72.4±0.9 | 74.8±1.2 | 71.9 |
| CoCo | 72.1±1.0 | 72.8±0.9 | 71.3±1.2 | 70.6±1.1 | 73.3±0.9 | 74.6±0.8 | 72.5 |
| CoCA | **77.6±1.1** | **78.3±1.3** | **74.1±1.2** | **69.5±1.1** | **75.7±1.3** | **77.1±1.0** | **75.4** |

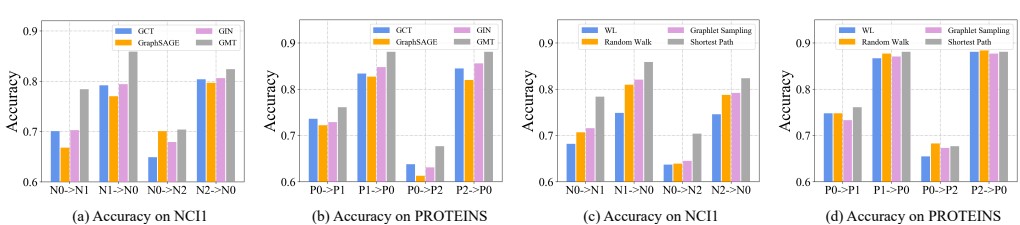

Figure 5: The performance with different GNNs and kernels on different datasets. (a), (b) are the performance of different GNNs, (c), (d) are the performance of different graph kernels.

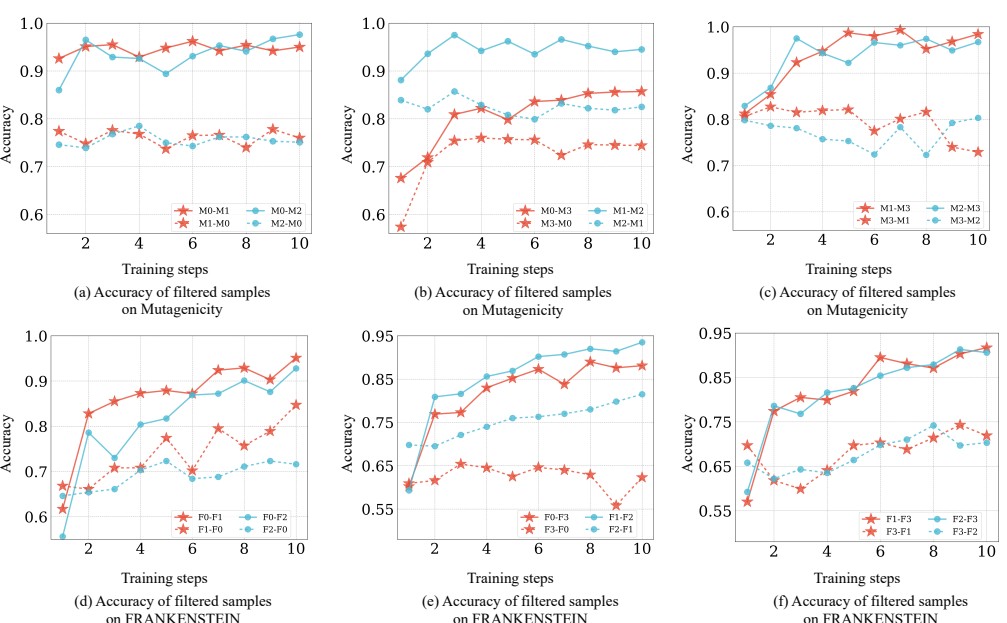

Figure 6: The accuracy of filtered samples on Mutagenicity and FRANKENSTEIN datasets. The solid line denotes the ratio of correct labels for samples filtered in MP branch, and the dotted line denotes the ratio of correct labels for samples filtered in SP branch.

Table 15: The classification results (in %) on Mutagenicity under edge density domain shift (source→target). M0, M1, M2, and M3 denote the sub-datasets partitioned with edge density. Bold results indicate the best performance.

| Methods | M0→M1 | M1→M0 | M0→M2 | M2→M0 | M0→M3 | M3→M0 | M1→M2 | M2→M1 | M1→M3 | M3→M1 | M2→M3 | M3→M2 | Avg. |
|---|---|---|---|---|---|---|---|---|---|---|---|---|---|
| GMT | 69.0±4.0 | 67.4±3.8 | 60.3±4.2 | 66.5±3.8 | 54.9±1.6 | 54.8±3.6 | 65.6±4.2 | 70.4±3.2 | 64.0±2.3 | 56.8±4.3 | 64.7±1.5 | 61.1±3.5 | 63.0 |
| DEAL | 75.6±0.8 | 72.0±1.1 | 68.4±1.6 | 72.0±1.1 | 58.3±0.8 | 65.6±1.8 | 76.7±0.9 | 79.3±0.6 | 65.2±1.3 | 72.3±1.5 | 69.4±1.1 | 75.8±1.2 | 70.9 |
| CoCo | 77.8±0.6 | 76.3±1.2 | 67.8±2.5 | 75.3±1.2 | 65.7±1.8 | 74.3±0.9 | 76.4±1.1 | 77.8±2.7 | 66.4±1.1 | 71.7±1.8 | 63.3±2.1 | 76.4±1.0 | 72.4 |
| CoCA | **82.4±1.5** | **80.8±1.2** | **74.5±1.7** | **79.6±2.1** | **74.8±2.2** | **79.2±0.7** | **83.4±0.9** | **85.7±0.6** | **73.9±0.8** | **81.3±1.5** | **77.8±0.7** | **83.3±1.4** | **79.7** |

