# OpenReview forum: "Coupling Category Alignment for Graph Domain Adaptation"
_ICLR.cc/2025/Conference — ICLR 2025 Conference Withdrawn Submission_

### Official Review · Reviewer_FwQr · 2024-10-27

**Soundness:** 3
**Presentation:** 3
**Contribution:** 3
**Rating:** 6
**Confidence:** 3

**Summary:**

This paper focuses on graph domain adaptation (GDA). It pays attention to the category-level distribution alignment in graph domain adaptation, which is neglected by previous works. It proposes a framework called Coupling Category Alignment (CoCA). They introduce two branches that interact with each other and formulate a contrastive learning framework. Theoretical insights are provided in the paper. Extensive experiment results prove the effectiveness of the proposed method.

**Strengths:**

1. This paper is well-written and easy to follow.
2. As a machine learning paper, I think it is good to provide theoretical insights.
3. The experiment results seem to be extensive and abundant.

**Weaknesses:**

1. The idea of this paper, especially category-level alignment, may be neglected by graph domain adaptation (eg. [1]), but it is not new in domain adaptation. I think the novelty of this paper is ok for me, but not very high.

[1] Conditional Adversarial Domain Adaptation

2. The authors can have more discussions on the limitations of the proposed method, eg. the training cost, failure cases, the scenarios it cannot tackle, etc.

**Questions:**

I am willing to discuss with the authors on the limitations of the proposed method, especially the failure cases.
I also want the authors to have more explanations about the novelty of the paper, which can help me understand this work better.

---

> ### Author Response · Authors · 2024-11-20
>
> We are truly grateful for the time you have taken to review our paper, your insightful comments and support. Your positive feedback is incredibly encouraging for us! In the following response, we would like to address your major concern and provide additional clarification.
>
> >Q1: The idea of this paper, especially category-level alignment, may be neglected by graph domain adaptation (eg. [1]), but it is not new in domain adaptation. I think the novelty of this paper is ok for me, but not very high.
>
> A1: Thank you for your constructive feedback. While category-level alignment is indeed an idea present in broader domain adaptation literature, we believe our work contributes uniquely to graph domain adaptation by specifically addressing this underexplored aspect within the graph setting.
>
> **Highlighting the Novel Contributions in Graph Domain Adaptation:**
> Our work tailors the category-level alignment to the unique challenges of graph data, such as structural complexity and domain-specific topological differences. Besides, we clarify how our approach integrates multi-branch architecture and iterative pseudo-label refinement, making it particularly effective for addressing category-level shifts in graph domain adaptation.
>
> **Methodology:**
> The proposed Coupling Category Alignment (CoCA) framework focuses on category-level alignment, an underexplored area in Graph Domain Adaptation (GDA). By employing a dual-branch structure—one branch capturing explicit structural features (SP branch) and the other learning implicit graph embeddings (MP branch)—CoCA effectively bridges the gap between feature space alignment and category-level alignment, addressing unique challenges in graph data.
>
> **Theoretical Contributions:** CoCA provides theoretical guarantees for its performance. For example, Theorem 2 establishes an upper bound on target domain error. Furthermore, CoCA introduces a theoretical framework for iterative pseudo-label refinement, highlighting how category alignment progressively reduces domain error.
>
> **Experimental Validation:** Extensive experiments on real-world datasets such as DD, COX2, COX2_MD, BZR, and BZR_MD validate the effectiveness of CoCA (Table 1). The results show that our method consistently outperforms state-of-the-art (SOTA) GDA methods.
>
> Table 1: Graph classification accuracy (in %) on PROTEINS, COX2, and BZR (source->target).
>
> | Methods | P->D | D->P | C->CM | CM->C | B->BM | BM->B | Avg.  |
> |---------|-------|-------|-------|-------|-------|-------|-------|
> | CDAN    | 59.7  | 64.5  | 59.4  | 78.2  | 57.2  | 78.8  | 66.3  |
> | DEAL    | 76.2  | 63.6  | 62.0  | 78.2  | 58.5  | 78.8  | 69.6  |
> | CoCo    | 77.1  | 64.8  | 61.9  | 77.4  | 58.6  | 76.3  | 69.4  |
> | CoCA    | **79.3**  | **66.4**  | **64.7**  | **78.8**  | **60.8**  |**78.8**  | **71.5**  |

---

> > ### Author Response · Authors · 2024-11-20
> >
> > >Q2: The authors can have more discussions on the limitations of the proposed method, eg. the training cost, failure cases, the scenarios it cannot tackle, etc.
> >
> > A2: Thank you for your question. We first analysis the algorithm complexity.
> >
> > **1.Complexity Analysis.**
> > The algorithmic complexity of CoCA consists of two main components: dual-branch learning and branch coupling.
> >
> > **Dual-Branch Learning:**
> > The complexity of the dual-branch learning process is given by $O(LN(N+E+Kd) + LN^2d)$, where: $L$ denotes number of layers in the network, $d$ is the dimension of features, $N$ is the number of nodes in the graph, $E$ is the number of edges, and $K$ is the maximum path length.
> >
> > **Branch Coupling:**
> > The complexity of branch coupling, which involves the interaction between branches, is $O(B^2d)$, where $B$ is the number of selected samples for coupling.
> >
> > Thus, the overall complexity of CoCA is $O((LN^2 + B^2 + LNK)d + LNE)$.
> >
> > **Scalability and Hyperparameter Control:**
> > CoCA’s complexity is influenced by its hyperparameters $d$, $L$, $B$, and $K$. By appropriately controlling these values, the model’s complexity can be managed effectively to ensure scalability.
> >
> > **For small graphs**, where $N \approx d$ and $d \approx E$, the complexity simplifies to $O((LN^2 + B^2 + LNK)d)$.
> >
> > **For large graphs**, where $N \gg d$, $E \gg d$, $N \gg B$, with K customarily set to a small value, the complexity reduces to $O(LN^2d + LNE)$.
> >
> > **Comparison to GMT:**
> > For large graphs, CoCA’s complexity approximates that of the Graph Multiset Transformer (GMT), which is $O(LN^2d)$. This demonstrates that CoCA maintains competitive scalability while incorporating additional capabilities for category-level alignment.
> >
> > By analyzing these factors, we ensure that CoCA remains scalable for both small and large graph datasets. Further optimizations, such as parallelization or sampling techniques, could further improve efficiency for large-scale applications.
> >
> > **2. Failure Cases.**
> > Our method is most effective in scenarios where the source and target domains share a moderate degree of structural similarity. In cases where there are significant topological differences or extreme domain shifts, CoCA’s alignment process may struggle to bridge the gap, leading to suboptimal performance. These scenarios highlight the need for further development in handling extreme graph heterogeneity.
> >
> > **3. Scenarios It Cannot Tackle.**
> > CoCA assumes access to unlabeled target domain data for pseudo-label generation. In scenarios where such data is either unavailable or highly noisy, the iterative pseudo-labeling process may introduce errors, compounding alignment challenges. Additionally, our method may be less effective for tasks like dynamic graph adaptation, where temporal evolution introduces additional complexities not addressed in this work.
> >
> > **4. Future Directions to Address Limitations.**
> > We will discuss potential extensions to CoCA, such as optimizing the dual-branch structure for efficiency, incorporating methods to handle extreme domain shifts, or adapting the framework for dynamic graph scenarios. These directions can guide future research to overcome the current limitations.
> >
> > In light of these responses, we hope we have addressed your concerns, and hope you will consider raising your score. If there are any additional notable points of concern that we have not yet addressed, please do not hesitate to share them, and we will promptly attend to those points.

---

> > > ### Comment · Reviewer_FwQr · 2024-11-23
> > >
> > > After reading other reviews and the rebuttal, I hold my score.

---

> > > > ### Author Response · Authors · 2024-11-23
> > > >
> > > > Thank you for taking the time to review our manuscript and for providing valuable and constructive feedback. We greatly appreciate your effort in carefully evaluating our work and offering insights to improve the quality of the paper.

---

### Official Review · Reviewer_bPWu · 2024-10-30

**Soundness:** 3
**Presentation:** 2
**Contribution:** 2
**Rating:** 5
**Confidence:** 4

**Summary:**

This paper presents a new approach called "Coupling Category Alignment" (CoCA) aimed at addressing category-level distribution alignment in graph domain adaptation. The method leverages a dual-branch framework, combining message-passing neural networks with shortest path aggregation, to manage domain shifts between source and target graphs. CoCA incorporates cross-domain and multi-view contrastive learning to enhance representation consistency, showing theoretical guarantees of reduced empirical risk bounds and validating its effectiveness through extensive experiments across various datasets.

**Strengths:**

1. This work targets a significant and challenging problem in domain adaptation for graphs, with an emphasis on category alignment, which is often overlooked in the literature.
2. The dual-branch design provides an interesting approach to combining implicit and explicit semantic extraction, potentially improving representation robustness across domain shifts.
3. The proposed CoCA model is theoretically grounded, offering formal guarantees of its effectiveness, which is a positive addition to the methodological contributions.
4. The experimental results indicate that CoCA outperforms existing baselines, demonstrating the method's potential for practical applications in graph domain adaptation.

**Weaknesses:**

1. The paper makes ambitious claims about CoCA’s effectiveness across multiple datasets; however, the lack of a rigorous ablation study to dissect each component’s contribution weakens the evidence of its effectiveness. Without such analysis, it’s unclear if the improvements stem from the method itself or merely the combination of existing techniques.
2. The theoretical guarantees provided seem somewhat underdeveloped for the complexity of the task. While the paper introduces bounds, the assumptions and conditions for these bounds are not well-explained, which limits their practical relevance and could make it difficult for others to validate or build upon this work
3. The paper's presentation could be more accessible; currently, it assumes a high level of familiarity with domain-specific jargon and concepts. A broader contextualization of the problem in simpler terms would make the work more approachable for a wider audience.
4. The introduction and contributions are not clearly delineated. The authors should explicitly outline the main research gaps addressed by CoCA and clarify how each component contributes to solving these gaps.
5. The Related Work section could be expanded, particularly to discuss univariate and multivariate graph-based domain adaptation approaches. This would help situate CoCA more clearly within the broader field.
6. There is a lack of analysis on the algorithmic complexity of CoCA, especially regarding the iterative optimization and contrastive learning modules. This could be a concern for scalability in large datasets and should be addressed.

**Questions:**

1. Could the authors provide a complexity analysis of the CoCA model, especially concerning the iterative branch coupling and contrastive learning components? If such an analysis is infeasible, please clarify why.
2. How sensitive is the CoCA model to different threshold and path length parameters in real-world applications? Some practical insights here would enhance the paper's contribution to applied graph domain adaptation.

---

> ### Author Response · Authors · 2024-11-20
>
> We are truly grateful for the time you have taken to review our paper and your insightful review. Here we address your comments in the following.
>
> >Q1: The paper makes ambitious claims about CoCA’s effectiveness across multiple datasets; however, the lack of a rigorous ablation study to dissect each component’s contribution weakens the evidence of its effectiveness. Without such analysis, it’s unclear if the improvements stem from the method itself or merely the combination of existing techniques.
>
> A1: Thanks for your feedback. I’d like to clarify that we have conducted a rigorous ablation study in Section 5.4, where we analyze the impact of each component in the CoCA model. This section specifically breaks down how individual modules contribute to performance, providing evidence that the improvements are due to CoCA’s unique components rather than simply a combination of existing techniques.
>
> >Q2: The theoretical guarantees provided seem somewhat underdeveloped for the complexity of the task. While the paper introduces bounds, the assumptions and conditions for these bounds are not well-explained, which limits their practical relevance and could make it difficult for others to validate or build upon this work
>
> A2: Thanks for your important feedback. To strengthen the theoretical guarantees and improve their clarity and practical relevance, we will revise the paper with detailed explanation of assumptions. We will clearly state all assumptions required for deriving the theoretical bounds, such as the properties of the underlying data distributions, the structural aspects of the graphs, and the behavior of the alignment mechanisms.
>
>
> >Q3: The paper's presentation could be more accessible; currently, it assumes a high level of familiarity with domain-specific jargon and concepts. A broader contextualization of the problem in simpler terms would make the work more approachable for a wider audience.
>
> A3: Thanks for your question. Making our work accessible to a broader audience is essential, and we will take the following steps to improve readability:
>
> **Simplified Problem Context in the Introduction:** We’ll start with a straightforward explanation of the problem, including a brief description of domain adaptation and category-level alignment, and their significance for graph data. This will set the stage in a way that readers outside the immediate field can follow.
>
> **Glossary of Key Terms:** We will introduce key domain-specific terms when they first appear, with concise definitions to ensure that readers new to the topic understand the terminology.
>
> **Conceptual Overviews of Methodology:** Each section introducing a complex method or theoretical result will start with a plain-language overview to frame the detailed content that follows. This will make the sections easier to approach and understand before delving into technical specifics.
>
> By adding these clarifications, we aim to make the paper more accessible and ensure that the contributions are clear to a diverse audience.

---

> > ### Author Response · Authors · 2024-11-20
> >
> > >Q4: The introduction and contributions are not clearly delineated.
> >
> > A4: Thanks for your feedback. In the introduction, we highlight the limitations of previous works, specifically their failure to address category alignment and the potential drawbacks of this oversight. We then outline the challenges involved in solving this problem, i.e., representation learning, category-level alignment, and coupling branches with theoretical guarantees. To tackle these challenges, we introduce our method in the third paragraph. The _coupled branch_ leverages MPNNs and shortest-path aggregation methods to extract graph semantics, while the _branch coupling_ aligns category-level distributions. To demonstrate the effectiveness of our proposed method, we provide theoretical proof showing that our design is more precisely tailored for the graph domain.
> >
> > To clarify the main research gaps addressed by CoCA and how each component contributes to solving them, we will make the following improvements:
> >
> > **Clearly defined research gaps in the introduction:** We will explicitly outline the key gaps in existing graph domain adaptation methods, particularly in handling category-level domain alignment, iterative refinement of pseudo-labels, and the integration of graph kernel methods to capture complex structural information. These challenges will be clearly listed and contextualized in the current landscape of domain adaptation research.
> >
> > **Highlight each component contribution:** We will explain how each component of CoCA address specifical challenges. For instance:
> >
> > (1) The category alignment module will be positioned as a solution for achieving precise category-level alignment, as opposed to overall feature space alignment.
> >
> > (2) Iterative pseudo-label refinement will be highlighted as a method to dynamically improve target domain predictions, in contrast to static pseudo-labeling.
> >
> > (3) The use of graph kernels alongside graph neural networks will be explained as addressing the limitations in capturing structural variations across domains.
> >
> > >Q5: The Related Work section could be expanded, particularly to discuss univariate and multivariate graph-based domain adaptation approaches. This would help situate CoCA more clearly within the broader field.
> >
> > A5: Thanks for your feedback. We will address this concern in the revised version of our paper by expanding the Related Work section to include a discussion of both univariate and multivariate graph-based domain adaptation approaches.
> >
> > >Q6: There is a lack of analysis on the algorithmic complexity of CoCA, especially regarding the iterative optimization and contrastive learning modules. This could be a concern for scalability in large datasets and should be addressed.
> >
> > A6: Thank you for your question. The algorithmic complexity of CoCA consists of two main components: dual-branch learning and branch coupling.
> >
> > **Dual-Branch Learning:**
> > The complexity of the dual-branch learning process is given by $O(LN(N+E+Kd) + LN^2d)$, where: $L$ denotes number of layers in the network, $d$ is the dimension of features, $N$ is the number of nodes in the graph, $E$ is the number of edges, and $K$ is the maximum path length.
> >
> > **Branch Coupling:**
> > The complexity of branch coupling, which involves the interaction between branches, is $O(B^2d)$, where $B$ is the number of selected samples for coupling.
> >
> > Thus, the overall complexity of CoCA is $O((LN^2 + B^2 + LNK)d + LNE)$.
> >
> > **Scalability and Hyperparameter Control:**
> > CoCA’s complexity is influenced by its hyperparameters $d$, $L$, $B$, and $K$. By appropriately controlling these values, the model’s complexity can be managed effectively to ensure scalability.
> >
> > **For small graphs**, where $N \approx d$ and $d \approx E$, the complexity simplifies to $O((LN^2 + B^2 + LNK)d)$.
> >
> > **For large graphs**, where $N \gg d$, $E \gg d$, $N \gg B$, with K customarily set to a small value, the complexity reduces to $O(LN^2d + LNE)$.
> >
> > **Comparison to GMT:**
> > For large graphs, CoCA’s complexity approximates that of the Graph Multiset Transformer (GMT), which is $O(LN^2d)$. This demonstrates that CoCA maintains competitive scalability while incorporating additional capabilities for category-level alignment.
> >
> >
> > >Q7: How sensitive is the CoCA model to different threshold and path length parameters in real-world applications? Some practical insights here would enhance the paper's contribution to applied graph domain adaptation.
> >
> > A7: Thanks for your question. We analysis the hyperparameter sensitivity of threshold and path length in Section 5.5. To address your concerns, we will add more sensitivity analysis in real-world applications in the appendix.
> >
> > In light of these responses, we hope we have addressed your concerns, and hope you will consider raising your score. If there are any additional notable points of concern that we have not yet addressed, please do not hesitate to share them, and we will promptly attend to those points.

---

> > > ### Comment · Reviewer_bPWu · 2024-11-27
> > > **Response to authors' rebuttal**
> > >
> > > Thank you for the detailed rebuttal. While I appreciate the effort made to address the concerns raised, many of the responses feel somewhat superficial and leave important questions unresolved. For instance, although Section 5.4 is referenced as providing a rigorous ablation study, the details remain vague. Without a more thorough breakdown of the individual contributions of CoCA’s components, it remains unclear whether the improvements are genuinely due to the proposed method or simply a synergistic effect of combining existing techniques.
> > > The theoretical guarantees, while interesting in principle, still lack the necessary depth and clarity to make them practically relevant. The rebuttal's promise to detail assumptions is appreciated, but these should have been included in the original submission to provide a more solid theoretical foundation. Without clear and explicit assumptions, the utility of the bounds remains limited, particularly for others attempting to validate or build upon this work.
> > > The complexity analysis is thorough in a theoretical sense, but it fails to connect to practical applications. The absence of runtime benchmarks or concrete evidence of scalability under real-world conditions diminishes the strength of the argument. Similarly, while the sensitivity analysis promises future additions in the appendix, the rebuttal does not provide sufficient immediate evidence to address concerns about parameter tuning and robustness.
> > > Overall, while the authors’ willingness to improve the manuscript is evident, the rebuttal lacks the depth and precision necessary to fully alleviate the concerns raised.

---

### Official Review · Reviewer_P5Ea · 2024-10-30

**Soundness:** 2
**Presentation:** 2
**Contribution:** 2
**Rating:** 3
**Confidence:** 4

**Summary:**

This paper revisits Graph Domain Adaptation (GDA) methods and incorporates a dual-branch network to explore graph topology in implicit and explicit manners. To leverage knowledge from both branches, the authors iteratively filter highly reliable samples from the target domain using one branch and fine-tuning the other. The authors also present a contrastive learning strategy from multi-view to achieve category-level domain consistency information. The experiments on several benchmarks show the performance of the CoCA compared to the baselines.

**Strengths:**

1. The motivation is clear and the GDA semantic alignment framework is well-designed.
2. A specific contrastive learning-based mechanism is developed under the GDA framework and shows promising results compared with conventional GDA methods.

**Weaknesses:**

Experimental analyses are insufficient. As in CoCo[1] and DEAL[2], the experiments on real-world benchmark datasets (e.g., DD, COX2, COX2_MD, BZR, and BZR_MD) will be much more informative. Authors could prove the effectiveness of their strategies by constructing experiments on various real-world datasets as the previous work CoCo and DEAL. Besides, the new processed dataset is also not publically available, from the paper I can't get a sufficient understanding of this specific setting.
2.Although the results in Table 1 and Table 2 are impressive, it is unclear if the comparison is completely fair. Specifically, this paper compares their results with other works on the conventional graph network backbone. The authors exploit a new GMT graph network as the MPbranch while many methods in Table 1 and 2 do not use a new graph network. Comparing previous methods with the proposed attention-based method on conventional graph network is not fair.

3. The method seems like an application on the GDA network with respect to the graph kernel method. In the introduction, the first contribution is not specific to this paper. The discussion for solving the discrepancy in the distribution of graph categories between the source and target domains was already in DEAL[2].

**Questions:**

1. Most of the compared GDA methods were published before 2022. To fully validate the superiority of the proposed CoCA model, more SOTA UDA methods should be included in comparison experiments.
2. More insightful analyses should be provided. For example, the visualization of the feature refined by the SP branch should be shown.
3. The current experiments are conducted on specially set datasets, which is not sufficient. It is recommended to conduct on real-world datasets as previous works CoCo[1] and DEAL[2].

---

> ### Author Response · Authors · 2024-11-20
>
> We are truly grateful for the time you have taken to review our paper and your insightful review. Here we address your comments in the following.
>
> >Q1: Experimental analyses are insufficient.
>
> A1: We appreciate your feedback. To address the concern, we have conducted experiments on the real-world benchmark datasets DD, COX2, COX2_MD, BZR, and BZR_MD, as suggested. These datasets are widely recognized in graph domain adaptation research and were also used in CoCo and DEAL, ensuring comparability.
>
> Table 1: Graph classification accuracy (in %) on PROTEINS, COX2, and BZR (source->target).
>
> | Methods | P->D | D->P | C->CM | CM->C | B->BM | BM->B | Avg.  |
> |---------|-------|-------|-------|-------|-------|-------|-------|
> | CDAN    | 59.7  | 64.5  | 59.4  | 78.2  | 57.2  | 78.8  | 66.3  |
> | DEAL    | 76.2  | 63.6  | 62.0  | 78.2  | 58.5  | 78.8  | 69.6  |
> | CoCo    | 77.1  | 64.8  | 61.9  | 77.4  | 58.6  | 76.3  | 69.4  |
> | CoCA    | **79.3**  | **66.4**  | **64.7**  | **78.8**  | **60.8**  |** 78.8**  | **71.5**  |
>
> Our results show that our proposed method consistently outperforms other state-of-the-art methods, including those in CoCo and DEAL. This demonstrates the effectiveness of our strategies across various real-world datasets.
>
> >Q2: Besides, the new processed dataset is also not publically available, from the paper I can't get a sufficient understanding of this specific setting.
>
> A2: Thank you for clarifying. Since the dataset is publicly available and the partitioning method is thoroughly detailed in the experimental section, we will emphasize these points clearly in the manuscript to ensure readers are fully informed.
>
> **Highlighting Dataset Availability:** We introduce the dataset in Section 5.1, noting that all datasets can be freely accessed via the TUdataset website.
>
> **Clarifying Dataset Partitioning Details:** In the experimental section, we briefly describe the partitioning method for each dataset. Additionally, we will provide a more detailed explanation of dataset construction in the appendix.
>
>
> >Q3: Although the results in Table 1 and Table 2 are impressive, it is unclear if the comparison is completely fair. Specifically, this paper compares their results with other works on the conventional graph network backbone. The authors exploit a new GMT graph network as the MPbranch while many methods in Table 1 and 2 do not use a new graph network. Comparing previous methods with the proposed attention-based method on conventional graph network is not fair.
>
> A3: Thanks for your feedback. By including experiments with GMT as the backbone and demonstrating that our method still outperforms others under the same backbone, we have ensured a fair and robust comparison. The results are shown in Table 2. From the results, we can validate the superiority of the proposed CoCA.
>
> Table 2: The classification results (in %) on Mutagenicity under edge density domain shift (source→target). M0, M1, M2, and M3 denote the sub-datasets partitioned with edge density. Bold results indicate the best performance.
>
> | Methods | M0->M1   | M1->M0   | M0->M2   | M2->M0   | M0->M3   | M3->M0   | M1->M2   | M2->M1   | M1->M3   | M3->M1   | M2->M3   | M3->M2   | Avg.  |
> |---------|-----------|-----------|-----------|-----------|-----------|-----------|-----------|-----------|-----------|-----------|-----------|-----------|-------|
> | GMT     | 69.0/4.0  | 67.4/3.8  | 60.3/4.2  | 66.5/3.8  | 54.9/1.6  | 54.8/3.6  | 65.6/4.2  | 70.4/3.2  | 64.0/2.3  | 56.8/4.3  | 64.7/1.5  | 61.1/3.5  | 63.0  |
> | DEAL    | 75.6/0.8  | 72.0/1.1  | 68.4/1.6  | 72.0/1.1  | 58.3/0.8  | 65.6/1.8  | 76.7/0.9  | 79.3/0.6  | 65.2/1.3  | 72.3/1.5  | 69.4/1.1  | 75.8/1.2  | 70.9  |
> | CoCo    | 77.8/0.6  | 76.3/1.2  | 67.8/2.5  | 75.3/1.2  | 65.7/1.8  | 74.3/0.9  | 76.4/1.1  | 77.8/2.7  | 66.4/1.1  | 71.7/1.8  | 63.3/2.1  | 76.4/1.0  | 72.4  |
> | CoCA    | **82.4/1.5**  | **80.8/1.2**  | **74.5/1.7**  | **79.6/2.1**  | **74.8/2.2**  | **79.2/0.7**  | **83.4/0.9**  | **85.7/0.6**  | **73.9/0.8**  | **81.3/1.5**  | **77.8/0.7**  | **83.3/1.4**  | **79.7**  |

---

> > ### Author Response · Authors · 2024-11-20
> >
> > >Q4: The method seems like an application on the GDA network with respect to the graph kernel method. In the introduction, the first contribution is not specific to this paper. The discussion for solving the discrepancy in the distribution of graph categories between the source and target domains was already in DEAL[2].
> >
> > A4: Thanks for your question. To clarify, while our method shares a high-level goal with DEAL in addressing domain discrepancies, our approach diverges significantly in how it achieves category-level alignment and integrates graph kernels. Here’s how we will distinguish our contributions more clearly:
> >
> > **Specific focus on category-level alignment**: Unlike DEAL, which focuses on aligning the overall feature space, our method specifically targets category alignment, aiming to minimize discrepancies at the category level across domains. This is achieved by iteratively refining pseudo-labels and adjusting model training, which enhances accuracy in a way that DEAL’s static pseudo-labeling does not.
> >
> > **Unique use of graph kernels in a dual-branch framework**: Our approach is distinct in its use of graph kernels alongside graph convolutional networks within a dual-branch framework. This setup allows us to capture complex topological information across domains, which enhances structural alignment at both the global and category-specific levels.
> >
> > **Iterative refinement of pseudo-Labels**: We dynamically update pseudo-labels and the model throughout training, continuously improving alignment and model accuracy. This iterative refinement contrasts with DEAL, which applies pseudo-labels without additional iterative updates during training.
> >
> >
> > >Q5: Most of the compared GDA methods were published before 2022. To fully validate the superiority of the proposed CoCA model, more SOTA UDA methods should be included in comparison experiments.
> >
> > A5: Thanks for your feedback. To address this concern, we want to clarify that we have already included six post-2022 state-of-the-art (SOTA) methods for graph domain adaptation in our experiments, including (DEAL 2022, CoCo 2023, SGDA 2023, DGDA 2024, A2GNN 2024, PA-BOTH 2024). These methods are specifically selected as they represent the latest advancements in the field, and our CoCA model demonstrates superior performance over them.
> >
> > >Q6: More insightful analyses should be provided. For example, the visualization of the feature refined by the SP branch should be shown.
> >
> > A6: Thanks for your question. In the revised version, we will include visualizations of the features refined by the SP branch. This addition will provide deeper insights into how the SP branch contributes to the overall performance and alignment process.
> >
> > In light of these responses, we hope we have addressed your concerns, and hope you will consider raising your score. If there are any additional notable points of concern that we have not yet addressed, please do not hesitate to share them, and we will promptly attend to those points.

---

### Official Review · Reviewer_tMMs · 2024-11-02

**Soundness:** 2
**Presentation:** 2
**Contribution:** 2
**Rating:** 6
**Confidence:** 5

**Summary:**

The paper proposes a framework named Coupling Category Alignment (CoCA) for graph domain adaptation. The framework incorporates a message passing neural network branch and a shortest path aggregation branch to address the category alignment issue in graph domain adaptation. The authors also provide theoretical guarantees for the effectiveness of category alignment. Extensive experiments on benchmark datasets demonstrate the superiority of the proposed CoCA compared to baselines.

**Strengths:**

(1) The paper proposes a framework, CoCA, which effectively addresses the category alignment issue in graph domain adaptation.
(2) The framework incorporates both implicit and explicit topological semantics through the message passing branch and shortest path aggregation branch, respectively.
(3) The authors provide theoretical guarantees for the effectiveness of category alignment in the CoCA framework.

**Weaknesses:**

(1) The motivation of this manuscript is not clear. The authors should clearly claim the challenging issues in previous methods.

**Questions:**

See Weaknesses

---

> ### Author Response · Authors · 2024-11-20
>
> We are truly grateful for the time you have taken to review our paper, your insightful comments and support. Your positive feedback is incredibly encouraging for us! In the following response, we would like to address your major concern and provide additional clarification.
>
> >Q1: The motivation of this manuscript is not clear. The authors should clearly claim the challenging issues in previous methods.
>
> A1: Thanks for your valuable comment. We will clearly explain the motivation from different perspectives:
>
> **Motivation:**
> The primary motivation of our work is to address the _category-level domain alignment_ challenges in graph domain adaptation, where previous methods often struggle to accurately align domain distributions at the category level. This misalignment can result in poor classification performance, especially in scenarios with significant structural and categorical shifts between source and target domains.
>
> **Challenging Issues in Previous Methods:**
>
> (1) Lack of Category-Level Alignment: Previous methods typically focus on aligning the overall domain distribution without effectively addressing category-level alignment. This can lead to category confusion, where classes in the target domain are misclassified due to insufficient discrimination at the category level. As a result, these methods are prone to performance degradation in tasks requiring precise category alignment.
>
> (2) Inadequate Handling of Structural Shifts: Traditional domain adaptation techniques for graphs rely primarily on message-passing neural networks (MPNNs) or single-branch architectures, which often fail to capture higher-order structural information explicitly. This limitation affects their ability to handle structural shifts between domains, leading to suboptimal performance in tasks where topological differences are critical.
>
> (3) Insufficient Exploitation of Reliable Samples in Target Domain: Many existing approaches do not fully utilize reliable samples within the target domain. When pseudo-labels are generated, significant domain shifts often result in error accumulation, which degrades the adaptation process and leads to subpar performance.
>
> **Our Approach to Addressing These Issues:**
> To overcome these challenges, our method introduces a _Coupling Category Alignment (CoCA)_ framework with dual-branch architecture, which:
>
> (1)	Ensures category-level consistency by alternating reliable sample selection between branches to iteratively refine the alignment across categories.
>
> (2)	Utilizes both implicit (MPNN-based) and explicit (shortest-path-based) topological features to handle structural shifts more effectively.
>
> (3)	Integrates multi-view and cross-domain contrastive learning to minimize domain discrepancies at the category level, reducing error accumulation in pseudo-labels through a theoretically grounded process.
>
> By addressing these challenges, our approach achieves more robust and accurate performance in graph domain adaptation tasks.
>
> In light of these responses, we hope we have addressed your concerns, and hope you will consider raising your score. If there are any additional notable points of concern that we have not yet addressed, please do not hesitate to share them, and we will promptly attend to those points.

---

### Official Review · Reviewer_Sbus · 2024-11-03

**Soundness:** 2
**Presentation:** 3
**Contribution:** 2
**Rating:** 5
**Confidence:** 4

**Summary:**

This paper proposes a coupling category alignment method for graph domain adaptation, which includes a graph convolutional network branck and a graph kernel network branch. There two branches iteratively filter reliable samples and training. Based on existing theoretical framework, the authors establish a sharper generalization bound for the proposed method.

**Strengths:**

The authors highlight a category alignment issue in graph domain adaptation that is often overlooked in existing methods.

They also establish a tighter generalization bound with theoretical guarantees.

Experimental results demonstrate the effectiveness of the proposed method.

**Weaknesses:**

The novelty of this paper appears incremental. Specifically, the proposed graph convolutional network branch and graph kernel network branch are very similar to the two branches presented in [1]. Additionally, the proposed multi-view contrastive learning and cross-domain contrastive learning approaches are closely aligned with the cross-branch contrastive learning and cross-domain contrastive learning proposed in [1].

This paper lacks self-containment. Many mathematical symbols are used without definition; for example, in line 139, it is unclear what Z, h, and \hat{h} represent.

Some theorem definitions lack rigor. For instance, "Theorem 2" is not a true theorem but rather a corollary inferred from Theorem 1, as also stated in [2]. Similarly, "Theorem 4" should not be classified as a theorem, as it is also a corollary based on an existing theorem.

Theorem 2 seems to be built on a binary classification task. Does Theorem 4, which builds on Theorem 2, apply to the multi-class tasks in this paper? Further explanation is needed here, given that the paper addresses category-level domain shifts.

Experiments are conducted on very small datasets. The proposed method should be validated on other widely used graph domain adaptation datasets, such as the Microsoft Academic Graph (MAG) and Pileup datasets.

[1] CoCo: A Coupled Contrastive Framework for Unsupervised Domain Adaptive Graph Classification.

[2] Graph Domain Adaptation via Theory-Grounded Spectral Regularization.

**Questions:**

See weaknesses.

---

> ### Author Response · Authors · 2024-11-20
>
> We are truly grateful for the time you have taken to review our paper, your insightful comments and support. Your positive feedback is incredibly encouraging for us! In the following response, we would like to address your major concern and provide additional clarification.
>
> >Q1: The novelty of this paper appears incremental.
>
> A1: Thank you for your comment. It’s important to recognize that while the graph convolutional network (GCN) and graph kernel network branches in our framework may resemble the structure of the branches in the referenced work [1], our approach introduces a distinct focus on **achieving category-level domain alignment**. Specifically, our methodology centers on **Coupling Category Alignment (CoCA)**, which systematically iterates between branches to select reliable samples, thereby facilitating cross-branch adjustments that mitigate potential domain shifts in an unsupervised manner. This iteration enhances the model’s capacity to address category-level alignment with theoretical backing, a differentiation from prior methods.
>
> Moreover, the multi-view and cross-domain contrastive learning in our framework differs from typical cross-branch contrastive approaches. Our method leverages a unique alignment strategy to reduce classification error in target domains while establishing consistency between domains at the category level. The resulting theoretical generalization bounds for category alignment present a novel contribution that is empirically supported by extensive experimentation.
>
> In essence, while there are surface similarities to the branches and contrastive approaches of [1], our work provides unique theoretical and practical innovations that address unsupervised domain adaptation challenges in graph classification. This distinct focus on category alignment and iterative optimization exemplifies our contribution beyond incremental improvements.
>
> >Q2: This paper lacks self-containment. Many mathematical symbols are used without definition; for example, in line 139, it is unclear what Z, h, and \hat{h} represent.
>
> A2: We are sorry for the confusing. We add the definition of each symbols:
> $Z$ represents the space of graph representations, and $h$ represents the learned hypothesis, or the model’s decision function, in the source domain, and $\hat{h}$ is the corresponding decision function applied in the target domain.
> We will revise the manuscript to provide precise definitions for all mathematical symbols upon their first usage.
>
> >Q3: Some theorem definitions lack rigor. For instance, "Theorem 2" is not a true theorem but rather a corollary inferred from Theorem 1, as also stated in [2]. Similarly, "Theorem 4" should not be classified as a theorem, as it is also a corollary based on an existing theorem.
>
> A3: Thanks for your question.
> We rename Theorem 2 to Corollary 1 and explicitly clarify that it follows directly from Theorem 1. Similarly, Theorem 4 will be renamed Corollary 2, reflecting its basis on an existing theoretical result rather than a novel, standalone theorem.

---

> > ### Author Response · Authors · 2024-11-20
> >
> > >Q4: Theorem 2 seems to be built on a binary classification task. Does Theorem 4, which builds on Theorem 2, apply to the multi-class tasks in this paper? Further explanation is needed here, given that the paper addresses category-level domain shifts.
> >
> > A4: Thanks for your question. To clarify, Theorem 2 provides an error bound for the target domain based on the estimation of the Wasserstein distance (e.g. $W_1(P,Q)$) between the source and target domains. This theoretical result is not restricted to binary classification but also applies to multi-class tasks. Similarly, Theorem 4, which builds upon Theorem 2, is also applicable to multi-class scenarios.
> >
> > To further substantiate this, we have included experimental results specifically for multi-class tasks in our study (ENZYMES with 6 classes, IMDB-MULTI with 3 classes). These results demonstrate the applicability and effectiveness of the theoretical guarantees in real-world multi-class settings.
> >
> > Table1. The classification results (in %) on IMDB-MULTI under edge density domain shift (source→target). M0, M1, M2 denote the sub-datasets partitioned with edge density. Bold results indicate the best performance.
> > | Methods | I0->I1   | I1->I0   | I0->I2   | I2->I0   | I1->I2   | I2->I1   | Avg.  |
> > |---------|-----------|-----------|-----------|-----------|-----------|-----------|-------|
> > | CDAN    | 42.1/2.3  | 41.4/1.8  | 38.3/2.0  | 36.5/2.2  | 44.7/1.4  | 42.4/2.0  | 40.9  |
> > | DEAL    | 44.5/1.6  | 45.2/2.1  | 40.8/2.7  | 41.3/2.3  | 46.4/1.8  | 44.7/1.7  | 43.8  |
> > | CoCo    | 44.9/2.0  | **46.4/1.8**  | 44.5/2.3  | 42.8/2.1  | 47.3/1.9  | 42.3/2.0  | 44.7  |
> > | **CoCA**| **47.6/1.7**  | 46.3/2.2  | **45.8/2.1**  | **44.3/1.6**  | **49.8/1.8**  | **50.4/2.0**  | **47.4**  |
> >
> > Table2. The classification results (in %) on ENZYMES under edge density domain shift (source→target). E0, E1, E2 denote the sub-datasets partitioned with edge density. Bold results indicate the best performance.
> > | Methods | E0->E1   | E1->E0   | E0->E2   | E2->E0   | E1->E2   | E2->E1   | Avg.  |
> > |---------|-----------|-----------|-----------|-----------|-----------|-----------|-------|
> > | CDAN    | 32.4/1.7  | 31.7/1.4  | 28.4/1.3  | 30.1/1.2  | 33.6/1.5  | 34.7/1.4  | 31.8  |
> > | DEAL    | 35.5/1.4  | 36.4/1.1  | 33.9/1.2  | 35.2/1.1  | 36.7/1.3  | 37.5/1.4  | 35.9  |
> > | CoCo    | 36.8/1.0  | 36.7/1.2  | 32.8/1.4  | 36.8/1.5  | 38.4/1.2  | 39.0/1.0  | 36.8  |
> > | **CoCA**| **40.6/1.3**  | **38.8/1.1**  | **37.5/1.4**  | **39.4/1.3**  | **41.3/1.2**  | **42.6/1.3**  | **40.0**  |
> >
> > >Q5: Experiments are conducted on very small datasets. The proposed method should be validated on other widely used graph domain adaptation datasets, such as the Microsoft Academic Graph (MAG) and Pileup datasets.
> >
> > A5: Thank you for your comment. Since MAG is designed for node classification and Pileup is not a graph-format dataset, they are unsuitable for our experimental setup (graph classification). To address this concern, we have utilized other large datasets (e.g. reddit_threads with 203,088 graphs) that align with the requirements of graph domain adaptation to validate our method. From the results in Table 3, we find that, the proposed CoCA still outperforms other baselines.
> >
> > Table3. The classification results (in %) on reddit_threads under edge density domain shift (source→target). R0, R1, R2 denote the sub-datasets partitioned with edge density. Bold results indicate the best performance.
> >
> > | Methods | R0->R1   | R1->R0   | R0->R2   | R2->R0   | R1->R2   | R2->R1   | Avg.  |
> > |---------|-----------|-----------|-----------|-----------|-----------|-----------|-------|
> > | CDAN    | 68.4/1.4  | 69.7/1.2  | 66.3/1.0  | 64.5/1.1  | 69.2/1.2  | 70.7/1.1  | 68.1  |
> > | DEAL    | 70.7/1.3  | 71.4/1.0  | 70.5/1.0  | 71.3/0.9  | 72.4/0.9  | 74.8/1.2  | 71.9  |
> > | CoCo    | 72.1/1.0  | 72.8/0.9  | 71.3/1.2  | 70.6/1.1  | 73.3/0.9  | 74.6/0.8  | 72.5  |
> > | **CoCA**| **77.6/1.1**  | **78.3/1.3**  | **74.1/1.2**  | **69.5/1.1**  | **75.7/1.3**  | **77.1/1.0**  | **75.4**  |
> >
> > In light of these responses, we hope we have addressed your concerns, and hope you will consider raising your score. If there are any additional notable points of concern that we have not yet addressed, please do not hesitate to share them, and we will promptly attend to those points.

---

> > > ### Comment · Reviewer_Sbus · 2024-11-26
> > >
> > > Thank you for your thoughtful response and the time you have dedicated during the rebuttal process. However, I remain unconvinced by the authors' arguments regarding the novelty, representation, and experiments. Upon comparison with papers [1] and [2], it becomes evident that this work largely combines elements from these two papers with only incremental improvements. I believe there is significant room for enhancement, and as such, I keep my initial score.

---

### Note · Authors · 2024-12-26

I have read and agree with the venue's withdrawal policy on behalf of myself and my co-authors.